# Greedy Growing Enables High-Resolution Pixel-Based Diffusion Models

**Cristina N. Vasconcelos, Abdullah Rashwan, Austin Waters, Trevor Walker, Keyang Xu, Jimmy Yan, Rui Qian, Shixin Luo, Zarana Parekh, Andrew Bunner, Hongliang Fei, Roopal Garg, Mandy Guo, Ivana Kajic, Yeqing Li, Henna Nandwani, Jordi Pont-Tuset, Yasumasa Onoe, Sarah Rosston, Su Wang, Wenlei Zhou, Kevin Swersky, David J. Fleet, Jason M. Baldridge, Oliver Wang**
*Google*

**Reviewed on OpenReview:** *https://openreview.net/forum?id=GpULi1dAMm*

## Abstract

We address the long-standing problem of how to learn effective pixel-based image diffusion models at scale, introducing a remarkably simple greedy method for stable training of large-scale, high-resolution models without the needs for cascaded super-resolution components. The key insight stems from careful pre-training of core components, namely, those responsible for text-to-image alignment *vs.* high resolution rendering. We first demonstrate the benefits of scaling a *Shallow UNet*, with no down(up)-sampling enc(dec)oder. Scaling its deep core layers is shown to improve alignment, object structure, and composition. Building on this core model, we propose a greedy algorithm that grows the architecture into high resolution end-to-end models, while preserving the integrity of the pre-trained representation, stabilizing training, and reducing the need for large high-resolution datasets. This enables a single stage model capable of generating high-resolution images without the need of a super-resolution cascade. Our key results rely on public datasets and show that we are able to train non-cascaded models up to 8B parameters with no further regularization schemes. Vermeer, our full pipeline model trained with internal datasets to produce $1024 \times 1024$ images, without cascades, is preferred by 44.0% *vs.* 21.4% human evaluators over SDXL.

## 1 Introduction

Training large-scale *Pixel-Space text-to-image Diffusion Models* (*PSDM*) to generate high-resolution images has been challenging due to optimization instabilities arising when growing model size and/or target image resolution, and due to the increasing demand for computational resources and high resolution training corpora. The predominant alternatives include *cascaded models*, comprising a sequence of diffusion models each targeting a progressively higher resolution and trained independently (Ho et al., 2022a; Saharia et al., 2022a; Nichol et al., 2022), and *latent diffusion models* (LDMs), where generation is performed in a low-dimensional latent representation, from which high resolution images are generated via a pre-trained latent decoder (Rombach et al., 2022).

In the development of cascaded models, it is challenging to identify sources of quality degradation and distortion resulting from design decisions at specific stages of the model. One well-known issue of cascades is the distribution shift between training and inference, where inputs to super-resolution or decoder models during training are obtained by down-sampling or encoding training images, but during inference they are generated from other models, and hence may deviate from the training distribution. This can cause amplification of unnatural distortions produced by models early in the cascade. The generation of realistic small objects such as faces or hands is one such challenge that has been difficult to diagnose in such models.

Beyond image generation *per se*, diffusion models serve as image priors for myriad downstream tasks, including inverse problems (Jalal et al., 2021; Kadkhodaie & Simoncelli, 2021; Kawar et al., 2022; Song et al.,

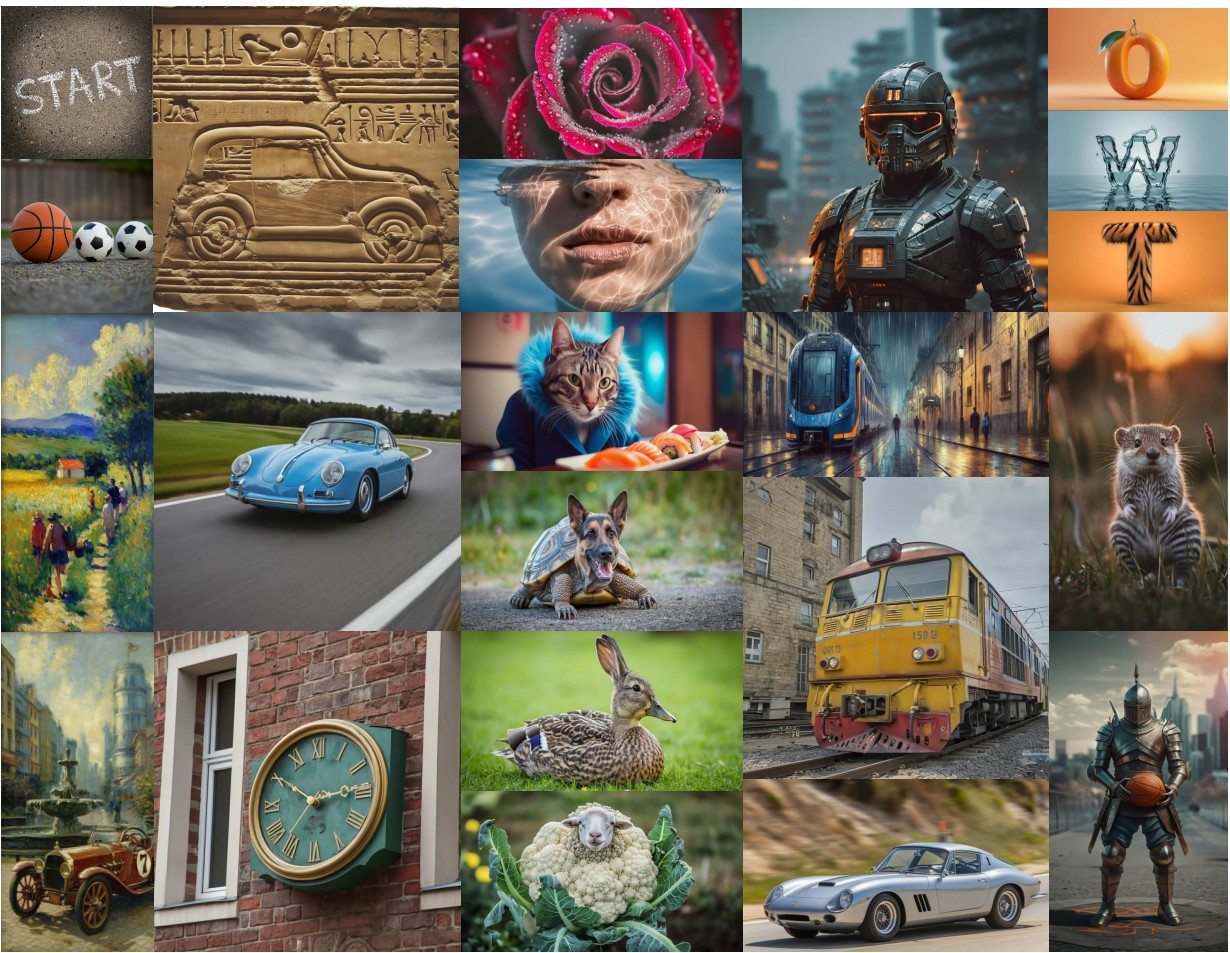

Figure 1: Images generated with our model Vermeer. (See Appendix A for the prompts.)

2023; Chung et al., 2023; Graikos et al., 2022; Tang et al., 2023; Jaini et al., 2023; Zhan et al., 2023; Song et al., 2024), or other generative tasks (Ho et al., 2022b; Levy et al., 2023; Poole et al., 2023; Tan et al., 2023; Bar-Tal et al., 2024; Chen et al., 2023; Tewari et al., 2023). Cascaded diffusion models are not readily applicable to such tasks, and as a consequence, many such applications rely solely on the score function from the base model of a cascade, often at a relatively low resolution. A high resolution end-to-end model would alleviate these issues, but model development and effective training procedures have been elusive.

Key barriers to training high resolution models include prohibitive resource requirements in both memory and computation. Existent recipes require large batch sizes during training to avoid instabilities, and as a consequence, intractably large amounts of memory for high-resolution images. Another issue concerns the need for high quality, high resolution training data. Existing training methods require large, diverse corpora of text-to-image pairs at the target resolution, while in practice, such data are not readily available at high resolution.

This paper introduces a framework for training high resolution, large-scale text-to-image diffusion models without the use of cascades. To that end we explore the extent to which one can decouple the training of 'visual concepts' associated with textual prompts, from the resolution at which one aims to render the image. Such disentanglement has two goals. It aims at a better understanding of alignment, composition and image fidelity (especially for well-known hard cases like generating consistent hands, text rendering, scene composition, etc.) as a function of model scaling (e.g., see Figure 3). Second, and of equal importance, our framework yields a robust and stable recipe for training large-scale, non-cascaded pixel-based models

targeting high-resolution generation. A bonus is that our recipe allows us to jointly train a single model with data comprising multiple resolutions, even if high-resolution text-image pairs are relatively scarce.

The contributions of this paper can be summarized as follows:

- We introduce a novel architecture, Shallow-UViT, which allows one to pretrain the *PSDM*'s core layers on datasets of text-image data (subsection 3.2), eliminating the need to train the entire model with high resolution images. This also allows us to investigate the emergent properties of PSDM representation scaling in isolation from layers targeting generation at the final resolution.

- We present a *greedy algorithm* for training the Shallow-UViT architecture that allows us to successfully train a high-resolution text-to-image model with small batch sizes (256 versus the typical 2k used in end-to-end solutions) (section 3).

- We show that one can significantly improve different image quality metrics by leveraging the representation pretrained at low-resolution, while growing model resolution in a greedy fashion. Scaling the core components of the Shallow-UViT architecture alone leads to significant improvements in image distribution, quality and text alignment (section 5).

- We demonstrate that these principles work at scale by presenting **Vermeer** (Figure 1), a model trained with our greedy algorithm on large-scale corpora, in conjunction with other well-known methods like asymmetric aspect ratio finetuning, prompt preemption and style tuning (section 6). Vermeer is shown to surpass previous cascaded and auto-regressive models across different metrics. In a human evaluation study with 500 challenging prompts and 25 annotators per image, Vermeer is preferred over SDXL (Podell et al., 2024) by a 2 to 1 margin.

## 2  Related work

Current high-resolution image generation with diffusion models presents a trade-off between architectural complexity and efficiency. Cascaded diffusion models (Nichol et al., 2022; Dhariwal & Nichol, 2021; Saharia et al., 2022b; Ramesh et al., 2022; Balaji et al., 2022) were originally introduced to circumvent the difficulty of training a single stage, end-to-end model. Cascaded models employ a multi-stage architecture that progressively up-scales lower-resolution images to address the computational challenges of generating high-resolution images directly. Nevertheless, they entail significant complexity and training overhead, as the stages of the cascade are trained independently.

Simple Diffusion (Hoogeboom et al., 2023b) sought to simplify the process by targeting the high resolution generation with a single stage model, introducing a novel UViT architecture and several useful modifications to training methods that improve stability. While this approach is shown to be effective, stability issues remain when targeting large-scale models, and high resolution images, due in part to their dependence on large batch sizes. In this work we adopt a similar UViT architecture, and some of their techniques for scaling, extending the model to much higher resolutions through greedy training. Through scaling the core backbone of the model, and with our greedy training procedure, we find with can scale to much high resolution models ($2\times$ to $8\times$ higher than Simple Diffusion), with excellent alignment, and much smaller batches when training high resolution layers of the model.

Another line of work proposed Matryoshka Diffusion Models (MDM) (Gu et al., 2023) that denoises multiple resolutions using a proposed Nested UNet architecture. They progressively train the network to preserve the representation at higher resolutions. We show in this work an alternate and simpler approach where denoising multiple resolutions is not required, but instead it is crucial to preserve the representation by freezing the pretrained weights as we grow the architecture up to its final design.

On another front, latent diffusion models (LDMs) (Rombach et al., 2022; Jabri et al., 2022; Betker et al., 2023) reduce computational costs by operating within a compressed latent representation. However, LDMs still require separate super-resolution or latent decoder networks to produce final high-resolution images.

The model we introduce also resembles progressive GAN training (Karras et al., 2018) in which layers of increasing resolution are added at each stage. Our work can be thought of as an extension of progressive growing for diffusion models, where we evaluate different growing configurations, and come up with a two-step recipe that arrives at a good trade-off of training efficiency, robustness, and generation quality. Specifically, while all layers remain trainable in progressive GANs, and a sequence of growing operations is performed before reaching the final architecture, we pretrain a core representation that remains frozen when training all grown layers at once up to the target resolution. We find that this is crucial to preserve the quality of the representation learned at lower resolutions.

## 3  Method

Our goal is to create a straightforward, stable methodology for training large scale pixel-space diffusion models that operate as a single stage model, i.e., non-cascaded, at inference time. To this end, we first revisit the UNet architecture, aiming to decouple layers that have a major impact on text-to-image alignment (*core components*) from those responsible for rendering at the target image resolution (*encoder-decoder* or *super-resolution components*). Next, we focus on pre-training the core components pretraining and on representation scaling (subsection 3.2). Finally, we present a greedy algorithm to grow the initial architecture core by adding encoder-decoder layers while protecting core layers' representation. This yields a single-stage model at inference time (subsection 3.3).

### 3.1  Text-to-image core components

UNet is the architecture of choice for diffusion models. Two architecture families are common. In one, convolutional networks comprise a stack of convolutional blocks alternated with pooling or downsampling layers in the encoder, and upsampling layers in the decoder. More recently, the UViT family emerged (Hoogeboom et al., 2023a), in which convolutional blocks are used at the higher layers of the encoder and decoder but augmented with transformer layers at the bottom of the UNet. In both architectural families, text conditioning is accomplished via cross-attention layers, also at the bottom, low-resolution layers of the UNet. In doing so, these layers are responsible for conditioning the models' deepest representation on the textual and/or multi-modal inputs. At these low-resolution layers, the text conditioning signal is able to influence the global image composition while the computational cost of attention is kept relatively low.

Our search for a methodology that allows stable training of large models starts by identifying and isolating *core layers* responsible for text-to-image alignment. Our main conjecture is that it is possible to reduce the instability typically observed during training large-scale PSDMs by warming up layers responsible for text-to-image alignment in isolation from layers responsible for target resolution encoding/decoding.

Specifically, we define the *core components* as those that directly interface with text conditioning signals and those that are crucial in the diffusion process. They can be described as:

- *Text encoding layers* combine one or more textual, character, and/or multimodal pretrained representations (such as those from Raffel et al. (2020b); Xue et al. (2022a); Liu et al. (2023); Radford et al. (2021a)), and project them into the embedding space of the UNet. Typically composed of MLP on top of pooling layers.

- *Core representation layers* comprise hidden layers in the main backbone interfacing with cross-attention layers. They include the bottom layers of the UNet architecture whose features are directly combined with the embedded text by the cross attention operation and layers between them.

- *Time encoding layers* map the diffusion time step into the model's embedding space. Typically designed as a sinusoidal positional encoder, followed by a shallow MLP. Despite not participating directly in the cross-attention operation, it is a core component of the diffusion process.

We isolate these core components of a *PSDM* text-to-image model in order to study their effect on the final model's properties. Next, we propose an architecture that enables the pretraining of these layers, and also supports the study of the properties emerging from scaling them.

## 3.2 Shallow-UViT

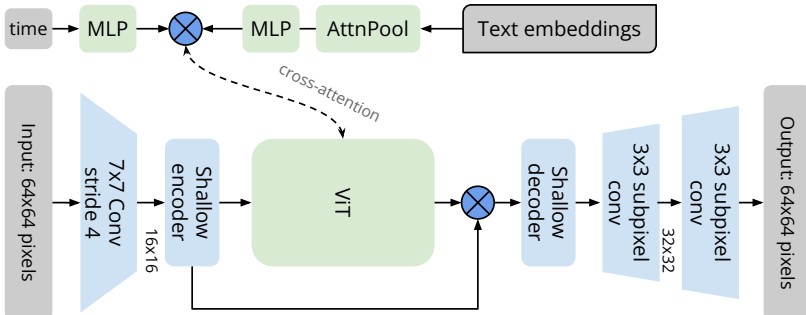

Figure 2: Shallow-UViT architecture: The input image grid is quickly reduced at the entry convolution, while a single residual block with no subsampling layers is used as a shallow encoder and decoder. The layers within the *core components* (in light green) are reused in the final end-to-end architecture, increasing its training stability, while remaining layers are discarded.

To assist the pretraining of the *core components* and, at the same time investigate the emerging properties from their scaling, we isolate the *core components* training and scaling from other confounding factors in the specification of the UNet's encoder-decoder layers. To that end, we simplify the UNet's conventional hierarchical structure, which operates on multiple resolutions, and define the Shallow-UViT (SU), a simplified architecture comprising a shallow encoder and decoder operating on a fixed spatial grid (Figure 2). Its encoder and decoder have a single residual block each, containing two layers of $3 \times 3$ convolutions with swish activations Ramachandran et al. (2017), and no upsampling or downsampling layers. As a result, they share the same spatial grid as the *core representation layers* at the bottom. The first convolutional layer at the entry of the architecture projects the input image into the fixed size grid used by its core layers. A corresponding upsampling head at the model's output reverses this operation. These input/output layers facilitate quickly projecting input images with larger resolution into the core representation with fixed and lower resolution.

As a second simplification, we restrict our investigation to the *core components* from the UViT model family owing to the uniform structure of its *core representation layers*. In contrast, the corresponding layers of convolutional UNets present a broader spectrum of design and hyperparameter choices, owing to their non-uniform yet hierarchical structure, rendering their analysis more complex.

An alternative to the proposed use of the Shallow-UViT architecture, might be to train the *core components* directly as an augmented ViT, as previously explored in latent diffusion models (Peebles & Xie, 2023). Our attempt to explore this approach proved not to be straightforward. A crucial difference between PSDM and LDM becomes highly relevant here. In the case of LDM, the transformer operates on latent tokens, and the diffusion model captures the latent token distribution. Our task, on the other hand, is to pretrain a rich representation directly from the raw pixels, for subsequent reuse as deep features within a higher-resolution pixel-space model. We conjecture that in such approaches the initial layers that are closer to the raw data do not transfer as well when reused within the final model.

Instead, our Shallow-UViT includes proxy additional layers that help with closing the gap between *core components* feature pretraining and their later use. That is, the auxiliary, yet shallow, input (output) and encoding (decoding) layers help adding expressiveness to the transformations between the input (output) and the models' hidden representation. Across the variations explored, the input convolution expands the number input channels up to 256 (we observed no improvement with more channels).

Beyond ablations on scaling (see section 5), we also found that certain variations for the Shallow-UViT composition tend to degrade performance in comparison to our best architecture. In particular, these include the removal of the shallow encoder/decoder blocks; the use of smaller/larger filters ($4 \times 4, 5 \times 5, .., 9 \times 9$) and strides (from 1 up to 8) at the entry convolution; and the use of a single output head with a subpixel

convolution upsampling by a factor of 4. We also experimented with convolutional *core representation layers*, but like Dosovitskiy et al. (2021), we find they under-perform their transformed-based counterparts.

### 3.3 Greedy growing

Here we describe a greedy approach to learn *PSDM*s for high-resolution images. Our process consists of two distinct stages, where we first pretrain the *core representation layers* at a low resolution using a Shallow-UViT architecture. Then, in the second phase, we replace the encoder/decoder layers with a more expressive set of UNet layers and train at the target resolution. This two-stage process is in contrast to progressive growing, which seeks to add one layer at a time. With this approach, we aim to mitigate the well-known instabilities observed during training of large models (Saharia et al., 2022b; Hoogeboom et al., 2023b), while making the best use of the available training corpora.

The *greedy growing* algorithm can be described as follows.

**Phase 1** In this phase, the *core components* of the chosen architecture are identified (see subsection 3.1), and a Shallow-UViT model is build on top of them. The Shallow-UViT is trained on the entire training collection of text-image pairs, as it is not limited to high resolution training images.

**Phase 2** The second phase greedily grows the Shallow-UViT's encoder/decoder (namely, throwing away the lower-resolution blocks and adding higher-resolution blocks) to obtain the final model. More specifically, this phase adds encoder and decoder layers at different resolutions, while preserving the *core representation layers* at the spatial resolution used during the first phase. In other words, the *core components* continue operating on a $16 \times 16$ grid. The added layers are randomly initialized, while the *core components* are initialized with the weights obtained on the first phase. The remaining components of the Shallow-UViT model are discarded.

Next, the grown model is trained. As it is a common practice for the generation of high fidelity images, at this point we filter the training data to remove text-image pairs with either image dimension is lower than the final model's target resolution. The *text encoding layers* and the *core representation layers* are kept frozen, to preserve the richness of the pretrained representation. The *time encoding layers*, on the other hand, are further tuned, jointly with the new encoder and decoder layers introduced in the second phase, which allows it to adapt to changes in the diffusion noise schedule. We adjusted the diffusion logSNR shift for high resolution images as suggested by Hoogeboom et al. (2023b), by a factor of $2\log(64/d)$. An optional third defrosting phase, may be applied in which all layers are jointly tuned, and seeks to benefit from the full capacity of the end-to-end architecture, but in practice we find that the first two phases are sufficient to obtain a good *PSDM*.

We empirically investigate the behaviour of the proposed algorithm in models of increasing size in subsection 5.2. We investigate the effects of splitting the training of the two tasks in phase one and phase two (i.e., for text-alignment and high-resolution generation), and we compare with models jointly trained from scratch, end-to-end. During these ablations, we constrain the greedy growing phase to use considerably smaller batch sizes than previous work, with no further regularization to demonstrate the optimization stability.

## 4 Experimental settings

**Shallow-UViT:** The proposed Shallow-UViT provides a proxy architecture for pre-training the *core components* of a larger PSDM. The ablation studies below us a specific instantiation of the model, but we expect Shallow-UViT to be flexible enough to be used with other component parts. In particular we adopt a combination of two pretrained text encoders for text conditioning: T5-XXL (Raffel et al., 2020a) with 128 sequence length and CLIP (VIT-H14) (Radford et al., 2021b) with 77 sequence length. Given a text prompt, we first tokenize and encode the text using the two encoders independently, and then concatenate the embeddings, yielding a final embedding with sequence length of 205. They are projected into model's *hidden size* by the *text encoding layers*. We keep the Shallow-UViT design fixed, except for changing the capacity by increasing

its width (hidden size) and depth (number of transformer's blocks), as detailed in Table 1. That produces a set of models varying from 672M up to 7.7B trainable parameters, mostly dedicated to the *core components*.

| model | transf. blocks | hidden size | MLP channels | heads | params[1] |
|---|---|---|---|---|---|
| Shallow-UViT Small | 6 | 1536 | 6144 | 12 | 672M |
| Shallow-UViT Large | 8 | 2048 | 8182 | 16 | 1.3B |
| Shallow-UViT Huge | 12 | 3072 | 12288 | 24 | 3.5B |
| Shallow-UViT XHuge | 16 | 4096 | 16384 | 32 | 7.7B |

Table 1: Shallow-UViT variants explored. Transformer layers operating at a $16 \times 16$ grid. The components within the shallow encoder and decoder block operate at same spatial resolution and hidden size.

| End-to-end model | channels per layer | residual blocks | *params |
|---|---|---|---|
| UViT Small | 256-384-768-1536 | 1-1-1-1 | 707M |
| UViT Large | 256-512-1024-2048 | 1-1-1-1 | 1.4B |
| UViT Huge | 384-768-1536-3072 | 1-1-1-1 | 3.6B |
| UViT XHuge | 512-1024-2048-4096 | 1-1-1-1 | 7.9B |

Table 2: Composition of the encoder-decoder layers grown on top of corresponding Shallow-UViT variants. *core components* identical to the corresponding shallow variant.

We stress that we do not claim that these specific *core components* are optimal. For instance, it is widely recognized that larger pretrained text encoders and longer token sequence lengths increase image quality (Saharia et al., 2022b; Balaji et al., 2022; Podell et al., 2024). Investigating the optimal design of each core component is beyond the scope of this work. Instead, the variations of the Shallow-UViT were intentionally designed to explore the performance benefits gained by increasing *core components*'s capacity independent of the remaining model components.

**Greedy growing:** In the experiments that follow we consider several different model sizes. Table 1 specifies the Shallow-UViT variants, while Table 2 specifies encoder/decoder parameterizations.

To ablate our hypothesis that greedy growing helps the model learn strong representations with larger, diverse corpora, we also train the full model on a high resolution subset of data used to train the Shallow-UViT; i.e., we simply removed all samples with resolution lower than the target model resolution. To that end, beyond greedy growing, we explore the three training baselines: 1) We create a baseline with all layers trained from scratch on this subset; 2) As an alternative to the frozen phase in the greedy growing, we fine-tune the *core components* on this smaller high resolution subset jointly with the grown components (randomly initialized); and 3) A third baseline adds the optional phase of unfreezing the *core components* after warming up the random weights for 500k steps. Models are trained for 2M steps in total.

The greedy growing algorithm aims to make training large-scale PSDMs at high resolutions more stable. In the case of Simple Diffusion (Hoogeboom et al., 2023b), large batch sizes and regularizers like dropout and multi-scale losses enable end-to-end training from scratch. To stress test the stability and convergence of our greedy growing algorithm, we restrict the batch size to 256 instead of the standard 2k, and we use no other explicit form of regularization. Under that restriction, our largest model (UVit-XHuge) presented numerical instabilities when trained from scratch or fine-tuned, as multiple numerical issues occurred during training. Thus, the results of this large model are presented only for the frozen, and freeze-unfreeze methods. This behaviour confirms observations in previous work and their need for large batch sizes.

**Dataset:** Rigorous evaluation of generative image models is challenging when models are trained on proprietary datasets. To avoid this issue, we first demonstrate our key findings through extensive empirical evaluations on a publicly available dataset, namely, Conceptual 12M (or CC12M) (Changpinyo et al., 2021).

---

[1]Number of trainable parameters after ignoring text encoders.

To evaluate the hypothesis that the greedy algorithm allows one to make good use of available corpora, we trained Shallow-UViT on the entire CC12M training set, while corresponding end-to-end models were trained with CC12M's subset of $8.7M$ images whose dimensions are equal or larger than 512 pixels. Those end-to-end models were therefore trained on 27.5% less data than the corresponding Shallow-UViT model. We do not explore more aggressive reduction of the corpora as the CC12M dataset is already a relatively small dataset for the models tested, and the variations tested already show overfitting characteristics under this setting, as discussed below. Thus, in what follows, the Shallow-UViT models were trained on $64 \times 64$ images, by resizing the smallest dimension of the images to 64 and random cropping along the remaining dimension as needed. The end-to-end models are trained at a target resolution of $512 \times 512$ as CC12M does not contain images at resolutions above 1024 pixels.

**Full pipeline model:** With those findings in place, we then explore the generation of larger images and train on a much larger curated datasets in order to show that the approach scales to state-of-the-art models (section 6). The resulting model, named Vermeer, is used to generate $1024 \times 1024$ images, well beyond the scale for which quantitative metrics are readily available. As such, with Vermeer we rely on human evaluation, in comparison to other recent models, like SDXL.

**Sampling:** Unless mentioned, the images and metrics were produced using 256 steps of a DDPM sampler (Ho et al., 2020) with classifier-free guidance (Ho & Salimans, 2021). We tune the guidance hyper-parameter by a FD-Dino/Clip (VIT-L14) trade-off as described in subsection 5.3.

## 4.1 Metrics

The evaluation of generative models poses considerable difficulties and constitutes an active research area (Kirstain et al., 2024; Xu et al., 2024; Hessel et al., 2021; Serra et al., 2023; Kim et al., 2024; Lee et al., 2023). In light of its inherent complexity, we utilize a multi-faceted evaluation strategy that combines image distribution metrics, text-alignment metrics and semantic question and answering metrics to validate our intermediary results, but the overall performance of our final model evaluation, Vermeer, is delegated to human evaluators (subsection 6.2). The following criteria are considered:

**Image distribution metrics:** We evaluate models on three key metrics, namely, the Fréchet Inception Distance (FID) (Heusel et al., 2017), the Fréchet Distance on Dino-v2 feature space (FD-Dino) (Stein et al., 2023; Oquab et al., 2023) and the Clip Maximum Mean Discrepancy (CMMD) distance (Jayasumana et al., 2023). FID is widely used to assess generative image models and select model hyper-parameters, but our findings corroborate its known limitations: it fails to reflect model improvements through training, it does not capture readily apparent distortions in individual images, and it does not correlate well with human perception (Stein et al., 2023; Otani et al., 2023; Jayasumana et al., 2023). Thus, in our study, we do not select training or sampling hyper-parameters solely on the basis of FID but, as described in Appendix 5.3, we review the trade-offs between the observed set of metrics.

We also note that metrics derived from image features vary considerably with image resolution. In what follows we compute metrics using the same resolution as the reference papers. The exception is for CMMD on Shallow-UViT outputs; the original metric taken at $336 \times 336$ pixels is dominated by up-sampling effects, obscuring differences between models. Thus, we replaced the original $ViT - L14$ operating at $336 \times 336$ by its version at $224 \times 224$ pixels.

**Multimodal metrics:** We adopt CLIP Score as a metric for text-image alignment, as it is widely used, and it complements image distribution metrics above, reflecting the consistency of the generated image with the given prompt. Unlike the original formulation based on ViT-B with path size 32 (Hessel et al., 2021) and previous papers in the area Saharia et al. (2022a); Hoogeboom et al. (2023b), we adopt the ViT-L (patch 14) embedding due to its improved representation. This choice results in lower absolute values of our CLIP Scores compared to previous results, however we noticed that these scores better correlate with the presence of absence of observed distortions.

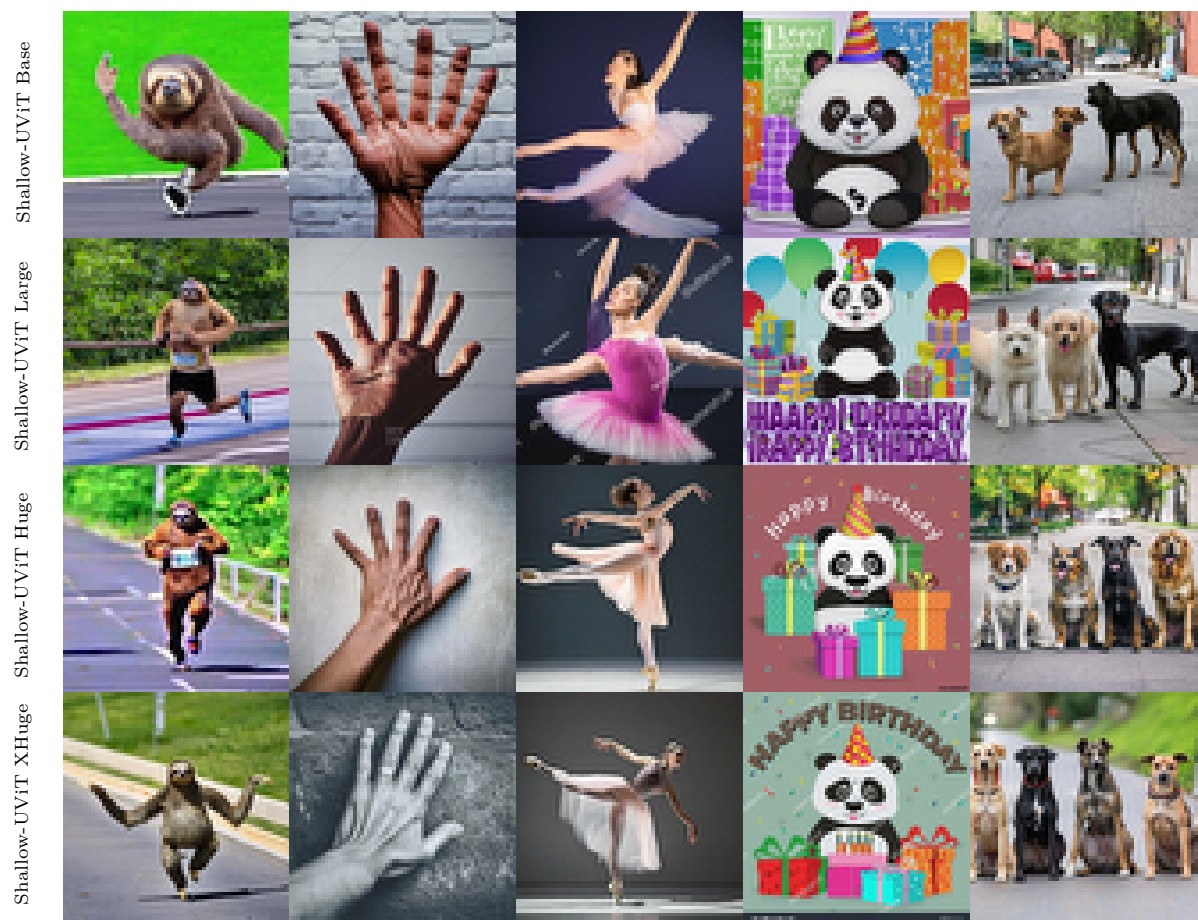

Figure 3: **Qualitative comparison of models with *core components* of increasing size** – Shallow-UViTs trained at $64 \times 64$ pixels using CC12M dataset only. Prompts: *'A sloth running a marathon, surprisingly outrunning all competitors.', 'A hand spread out on a wall. DSLR photograph.', 'Close-up portrait of a ballerina in mid-performance, with high motion and dramatic lighting.', 'Word art of "happy birthday", with a smiling panda wearing a party hat, surrounded by gift boxes and a birthday cake.', 'Four dogs on the street.'*

**Semantic QG/A frameworks:** One can also automatically generate question-answer pairs with a language model, and then compute image faithfulness by checking whether existing VQA models can answer the questions from the generated image (Hu et al., 2023; Cho et al., 2024). They were intended to address the shortcomings of existing metrics. Despite their effectiveness in evaluating color and material aspects, they often struggle in assessing counting, spatial relationships, and compositions with multiple objects. Such evaluation measures are naturally dependent on the quality of the underlying question generation (QG) and answering (QA) models. Here we adopt DSG (image-text alignment metric) and its set of $1k$ prompts (Cho et al., 2024). The DSG-1k test-prompts cover different challenges (e.g., counting correctly, correct color/shape/text rendering, etc.), semantic categories, and writing styles. A description of the QG, QA used, with qualitative and detailed results, are included in Appendix B.

# 5 Experiments

## 5.1 Pretraining and scaling the *core components*

We next use Shallow-UViT as a proxy architecture to investigate the effect of scaling PSDM's *core components*. We train Shallow-UViT variants on $64 \times 64$ images from the CC12M dataset for 2k steps. Image

| models$_{@64\times64}$ | FID$_{30k}$ ↓ | FD-Dino$_{30k}$ ↓ | CMMD$_{30k}$ ↓ | CLIP$_{score}$ ↑ |
|---|---|---|---|---|
| Shallow-UViT Base | 16.97 | 356.25 | 0.197 | 0.234 |
| Shallow-UViT Large | 14.80 | 236.24 | 0.156 | 0.240 |
| Shallow-UViT Huge | 8.81 | 133.51 | 0.139 | 0.244 |
| Shallow-UViT XHuge | **8.41** | **116.83** | **0.136** | **0.246** |

Table 3: Shallow-UViT variants with *core components* of increasing size trained on CC12M at resolution $64 \times 64$: Image distribution metrics evaluated on $30k$ samples from MSCOCO captions dataset. Scaling induces performace improvements on image distribution (FID, FD-Dino, CMMD) and text-image alignment ($CLIP_{score}$) metrics simultaneously.

| | DSG - VqVa Question Types | | | | |
|---|---|---|---|---|---|
| | Entities | Relations | Attributes | Global | DSG(↑) |
| #questions: | 3378 | 1485 | 1722 | 649 | |
| Shallow-UViT Small | 54.38 | 33.32 | 43.70 | 39.98 | 48.08 |
| Shallow-UViT Large | 59.93 | 39.36 | 48.75 | 43.68 | 52.54 |
| Shallow-UViT Huge | 69.18 | 48.52 | 54.36 | 43.30 | 60.25 |
| Shallow-UViT XHuge | **70.66** | **51.61** | **57.38** | **44.14** | **61.91** |

Table 4: Shallow-UVIT evaluated on $1k$ samples from DSG-1k dataset. Scaling *core components* improves performance across all semantic categories. Fine-grained results in Appendix B

distribution metrics and Clip-Score are obtained using $30k$ prompts from the MSCOCO-captions validation set (Chen et al., 2015), while the semantic metrics are extracted on the 1k prompts from DSG-1k (Cho et al., 2024). A summary of the impact of scaling the Shallow-UViT model is given in Tables 3 and 4, while fine grained results on semantic categories are reported in Appendix B. All performance measures indicate significant improvements due to model scaling. A smaller numerical gain is observed in the comparison of the larger two models, but the difference is reflected in qualitative comparisons of the models below.

Figure 3, presents a qualitative comparison of the results the Shallow-UViT variants on challenging prompts. They illustrate the impact of scaling on objects structure, composition and alignment (e.g., with numbers of objects depicted). Despite of the small training dataset, the larger models show significant improvement in generating intricate shapes like hands, body parts and text.

We observed further quantitative improvements across the metrics when training our larger models for longer (Shallow-UViT-Huge and Shallow-UViT-XHuge), but longer training also exhibits overfitting to the CC12 training samples. Figure 4 illustrates images generated using the Shallow-UViT XHuge model with increasing numbers of training steps. As training progresses, the model diverges from the original prompt to produce images that are closer to training samples from the CC12M dataset, and/or representing parts of the prompt only. This hidden phenomena was not associated with changes in the adopted metrics. We conjecture that this effect is largely aggravated by the small size of the training dataset.

Considering the complexity associated with evaluating improvements in representation and the limitations of automatic performance measures, we also ablate the effect of scaling the *core components* under a semantic task that is evaluated by human annotators. In this experiment we consider a *simple counting task*, defined here as the task of generating images of up to 5 objects based on a subset of text prompts from the numerical split of the Gecko benchmark (Wiles et al., 2024). We explore this task as a proxy for gauging both prompt consistency and the model's understanding of objects composition and shapes. It allows less subjective interpretation and noise in human judgments of the model's performance than other image qualities that are influenced by individual preferences. The task of counting under an open set would ultimately imply the ability to keep track of objects. Thus, this ablation emulates a much simpler version of the problem. Figure 5 shows the accuracy improvement associated with scaling observed over 59 prompts. Random condition uses a random number between 1-5. The detailed description of this experiment is presented on Appendix C.

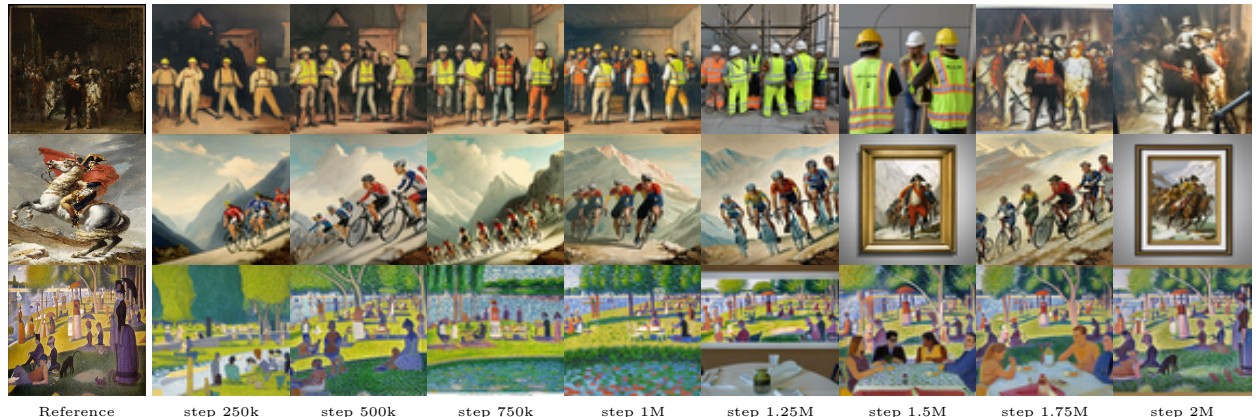

Reference    step 250k    step 500k    step 750k    step 1M    step 1.25M    step 1.5M    step 1.75M    step 2M

Figure 4: Overfitting and memorization of Shallow-UViT XHuge trained on CC12M. Prompts: (top) *A group of construction workers in the style of 'The Night Watch' by Rembrandt.*; (middle) *A dynamic rendition of a racing cyclist leading their team through a mountain pass, rendered in the style of 'Napoleon Crossing the Alps' by Jacques-Louis David.*; (bottom) *A group of friends enjoying a summer day at a riverside restaurant in the style of 'A Sunday Afternoon on the Island of La Grande Jatte' by Georges Seurat.*

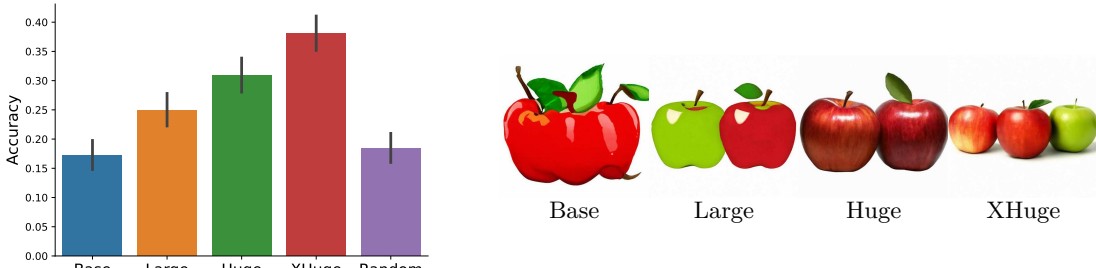

Figure 5: Measuring the impact of scaling on the counting task. Using 59 systematic prompts describing 1-5 objects. Five human annotators reviewed each image (95% bootstrapped confidence intervals are shown). Models with larger *core components* are observed to perform better on counting. Sample prompt: *3 apples.*

Given the shallow encoder-decoder structure of the Shallow-UViT architecture, we conjecture that the performance improvements observed here, on multiple metrics, are a direct consequence of scaling the *core components*. This hypothesis is further investigated via the reuse of their representation in the next section.

## 5.2 Experiments on Greedy growing

We next explore greedy growing of Shallow-UViT models to high resolution, non-cascaded models. We compare training models from scratch on the subset of the CC12M dataset filtered by the target resolution (512 pixels) with alternatives for reusing of the *core components* pretrained on the full dataset. They validate our main intuitions behind the greedy growing algorithm, i.e., that the introduction of new, untrained layers, as well as shifts in the distribution of the training data are known causes of the catastrophic forgetting phenomena Vasconcelos et al. (2022); Kuo et al. (2023); Yu et al. (2023) possibly damaging the pre-trained representation.

Tables 5 and 6 summarize performance as a function of model scale for greedy growing, along with various ablations of the training procedure. Our *greedy growing* recipe with frozen *core components*'s and its optional defrosting phase lead to the best results across the metrics. The optional defrosting phase is required for improving the performance of the smallest model ablated (UViT-Base). Its frozen counterpart showed signs of underfitting during training, as it has a small number of trainable parameters (217M) in the added layers. Under this low-capacity scenario, the defrosting phase offers a balance between protecting the *core components* representation and the use of the model's full capacity, as it reduces the degradation of the pretrained representation by warming up the growth layers. Other than this special case, the defrosting

phase did not appear to benefit larger models. These quantitative results agree with our hypothesis that the final model benefits from protecting the pretrained representation in our *greedy growing* algorithm.

Figure 6 qualitatively compares generations obtained by finetuning and freezing the *core components*. Additional qualitative comparisons are shown in Appendix D. They illustrate the the benefits of protecting the *core components* from the noise introduced when back-propagating through the randomly initialized growth layers. We observe that the low-resolution images produced by the use of the same representation under their original Shallow-UViT models produce objects whose shapes and parts are correctly defined.

The high-resolution images generated from early steps (20k) of finetuning the *core components* under the UVit architecture present objects with correct shapes superimposed with the diffusion noise. Soon after that (around 50k-100k steps) the quality of object shapes and structure decays as the training backpropagates the noise introduced by the growth layers through the pretrained representation.

Under the *greedy growing* regime and same number of training steps (20k steps) the frozen model is able to produce objects with correct shapes and parts, and maintain their composition as training progresses. Another direct side effect of maintaining the *core components* representation is the fast reduction of the diffusion noise early in training.

| model | | $tr.\ params$ | $steps$ | $\text{FID}_{30k}\downarrow$ | $\text{FD-Dino}_{30k}\downarrow$ | $\text{CMMD}_{30k}\downarrow$ | $\text{CLIP}_{score}\uparrow$ |
|---|---|---|---|---|---|---|---|
| UViT-Base | *scratch* | 707M | 2M | 27.90 | 624.34 | 1.355 | 0.241 |
| | *finetuning* | | 2M | 23.67 | 554.99 | 1.450 | 0.241 |
| | *frozen core* | 217M | 2M | 24.68 | 563.35 | 1.614 | 0.235 |
| | *freeze-unfreeze* | 217M/707M | 2M | **21.13** | **503.16** | **1.196** | **0.247** |
| UViT-Large | *scratch* | 1.4B | 2M | 21.73 | 498.82 | 1.156 | 0.247 |
| | *finetuning* | | 2M | 21.89 | 414.42 | 1.160 | 0.253 |
| | *frozen core* | 351M | 2M | **17.68** | **195.80** | **0.752** | **0.264** |
| | *freeze-unfreeze* | 351M/1.4B | 2M | 18.37 | 362.58 | 0.952 | 0.256 |
| UViT-Huge | *scratch* | 3.6B | 2M | 18.58 | 382.17 | 1.053 | 0.256 |
| | *finetuning* | | 2M | 17.52 | 302.28 | 0.988 | 0.264 |
| | *frozen core* | 723M | 2M | **15.21** | **156.24** | **0.663** | **0.268** |
| | *freeze-unfreeze* | 723M/3.6B | 2M | 16.17 | 231.94 | 0.683 | 0.262 |
| UViT-XHuge | *freeze* | 1.2B | 2M | **15.32** | **152.12** | **0.571** | **0.269** |
| | *freeze-unfreeze* | 1.2B/7.9B | 2M | 16.58 | 222.38 | 0.620 | 0.267 |

Table 5: End2end variants trained on CC12M dataset at $512 \times 512$ pixels and batch size 256: image distribution metrics (FID, FD-Dino and CMMD). Smaller models benefit from finetuning all their parameters. Larger models have more capacity in the encoder-decoder layers, and benefit from freezing the pretrained representations, under such a small batch size regime.

| model | | $steps$ | DSG - VqVa Question Types | | | | |
|---|---|---|---|---|---|---|---|
| | | | Entities | Relations | Attributes | Global | DSG |
| UVIT-Base | *scratch* | *2M* | 73.16 | 53.91 | 62.31 | 55.55 | 64.83 |
| | *finetuning* | *2M* | 70.23 | 49.90 | 58.89 | 53.24 | 62.75 |
| | *frozen* | *2M* | 69.57 | 49.36 | 58.22 | 53.39 | 61.16 |
| | *freeze-unfreeze* | *2M* | **73.40** | **53.54** | **62.83** | **56.86** | **66.13** |
| UVIT-Large | *scratch* | *2M* | 73.31 | 52.02 | 62.95 | 58.01 | 66.02 |
| | *finetuning* | *2M* | 75.01 | 54.11 | 65.82 | 57.86 | 67.39 |
| | *frozen* | *2M* | **78.97** | **61.55** | **67.19** | **61.40** | **72.13** |
| | *freeze-unfreeze* | *2M* | 74.67 | 55.45 | 64.08 | 58.78 | 67.79 |
| UViT-Huge | *scratch* | *2M* | 74.33 | 55.02 | 62.98 | 58.63 | 66.90 |
| | *finetuning* | *2M* | 77.29 | 56.40 | 67.13 | 62.56 | 69.67 |
| | *frozen* | *2M* | **82.59** | **64.65** | **70.35** | 61.86 | **75.15** |
| | *freeze-unfreeze* | *2M* | 79.04 | 58.11 | 65.97 | 60.86 | 71.50 |
| UViT-XHuge | *frozen* | *2M* | **83.70** | **66.77** | **70.01** | **62.94** | **75.70** |
| | *freeze-unfreeze* | *2M* | 81.14 | 60.44 | 69.40 | 60.25 | 73.53 |

Table 6: E2e variants at $512 \times 512$ pixels trained on CC12M dataset. Metrics evaluated on $1k$ samples from DSG-1k dataset. *DSG* results are aggregated across semantic categories. Fine-grained results in Appendix B.

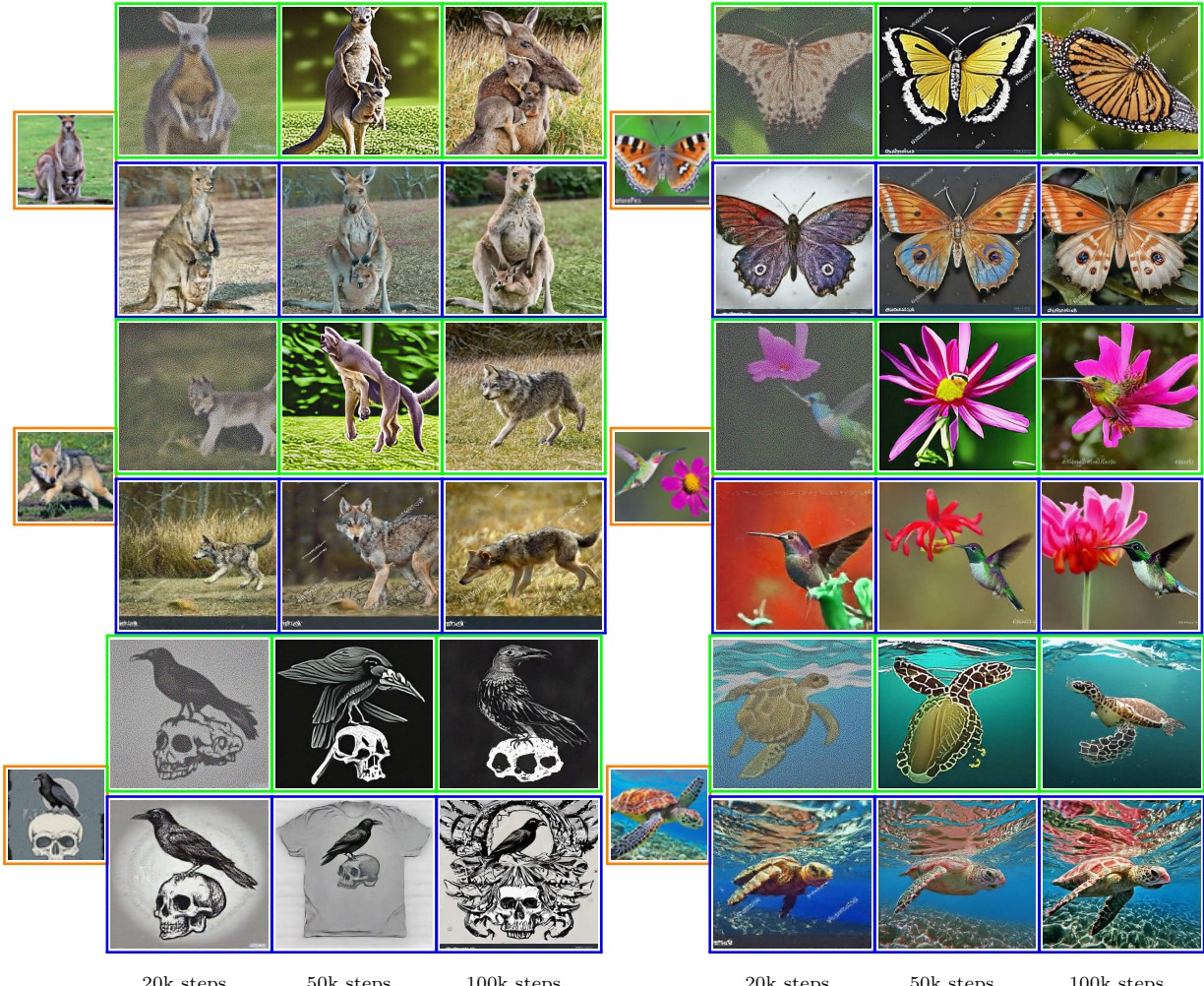

Figure 6: On catastrophic forgetting during early steps of finetuning: the pretrained representation quickly deteriorates due to noise introduced by the random weights from newly added layers. (from left to right) $64 \times 64$ image produced by the pretrained Shallow-Unet-Huge; followed by $512 \times 512$ images (in green) produced at early steps of finetuning (ft.) the core representation in an E2e model; and (in blue) freezing the core layers. *Differences better observed zooming in.* Prompts: *'A loving mother kangaroo carrying her joey in her pouch.'*; *'A close-up portrait of a butterfly, revealing the intricate patterns and textures on its wings in exquisite detail'*; *'A playful wolf pup chasing its own tail'*; *'A graceful hummingbird hovering near a bright pink flower'*; *'A dark and gothic illustration of a raven perched on a skull'*; *'A determined sea turtle swimming against the ocean current'.*

## 5.3 Guidance tuning

Diffusion model hyper-parameters affect both training and sampling quality. It is a common practice to tune the sampler guidance weights using FID-CLIP$_{score}$ trade-off curves (Saharia et al., 2022a; Hoogeboom et al., 2023b; Podell et al., 2024). In doing so one aims to strike a balance between images quality (by minimizing FID) and alignment with the text prompt (maximizing the CLIP$_{score}$ score). That said, it is well known that FID does not correlate particularly well with human perception (Stein et al., 2023; Otani et al., 2023; Jayasumana et al., 2023), and large guidance weights are known to increase CLIP-Score but tend to produce over-sharpened, high-contrast images and unrealistic objects (Ho & Salimans, 2021; Saharia et al., 2022b). Due to such limitations, despite widespread use of FID-CLIP$_{score}$ scores for performance comparisons, in practice they are adopted as loose measure of performance, and guidance weights are typically set through qualitative inspection.

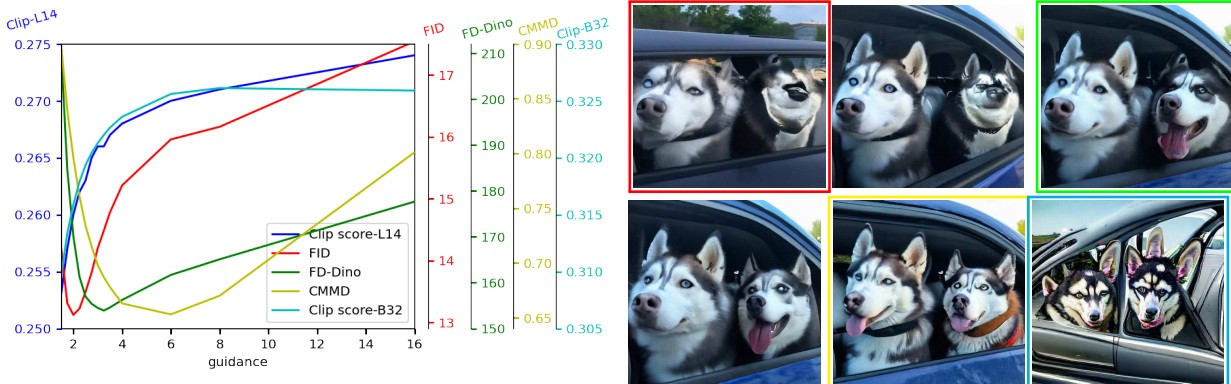

Figure 7: On the FID-CLIP tradeoff and the use of SOTA feature spaces for image and text-alignment distributions. (left) In the process of optimizing guidance, there are clear tradeoffs between different metrics (Fréchet distance and Maximum Mean Discrepancy) taken under different feature spaces (Inception, Dino, and Clip) used to represent the image distributions. (right) qualitative illustrations: sample images taken with increasing guidance values from left-to-right and top-to-bottom. A sample taken under the guidance value that minimizes FID is highlighted with a red bounding box, whereas those using guidance optimum values for minimizing FD-Dino and CMMD are distinguished with green and yellow highlights, respectively. In cyan: the saturation/cartoonish effect of increasing CLIP score further in detriment of the other metrics. *Differences better observed zooming in.* Prompt (from MSCOCO captions): *'Two huskies hanging out of the car windows.'*

Here we explore alternative metrics for hyper-parameters tuning, aiming to better reflect their deployment use, and ultimately human perception. These include recent measures with alternative feature spaces that exhibit better robustness in classification tasks, and align somewhat better with human judgements of image quality and alignment. More specifically, we investigate the use of FD-Dino and CMMD as alternatives to FID in the calibration of the guidance hyper-parameter. Figure 7 plots the response curve of different metrics as a function of guidance weight. They were measured using our UVIT-XHuge frozen model taken over 30k samples from the MSCOCO-caption validation set. It illustrates that the three image distribution metrics are minimized by very different guidance values. Similar curves are observed on the other models and training modalities, in which the best guidance value for minimizing FID, FD-Dino and CMMD are in increasing order. Figure 8 further illustrates samples obtained at the optimal values for each metric, and also when using the maximum guidance tested (16) for increasing CLIP$_{score}$ even further.

A qualitative analysis shows that by minimizing FID, one favors the generation of natural colors and textures, but under closer inspection, it fails to produce realistic object shapes and parts. We conjecture that this matches prior observations on the existence of texture vs shape bias by image classifiers (Geirhos et al., 2019). Guidance values minimizing Dino-v2 features, on the other hand, appear to produce natural color distributions and objects with natural shapes and composition. We adopt the value at this minimum as our new lower bound. Increasing guidance from that value tends to increase color-contrast and sharpening.

Images produced with guidance weights minimizing CMMD tend to produce images with initial signs of saturated colors and over-sharpening. Given its use of Clip features for image distribution comparison, this agrees with previous observations on CLIP$_{score}$. But unlike CLIP$_{score}$ curves, CMMD curves present an inflection point within the range investigated. We use this inflection point to define a closed range for our search of reasonable guidance weights. That is, the range of guidance weights between FD-Dino and CMMD minimums was observed to strike a balance between producing correct shapes and aesthetically pleasing images characterized by enhanced color contrast and sharp edges.

All results presented in this section have their image generated using guidance weights within the FD-Dino/CMMD trade-off range. The specific value selected was taken at the intersection of the optimal ranges of models under the same comparison. Following this approach, our Shallow-UViT results were obtained with guidance weights fixed at 1.75, and their corresponding UViT models with guidance 4.0.

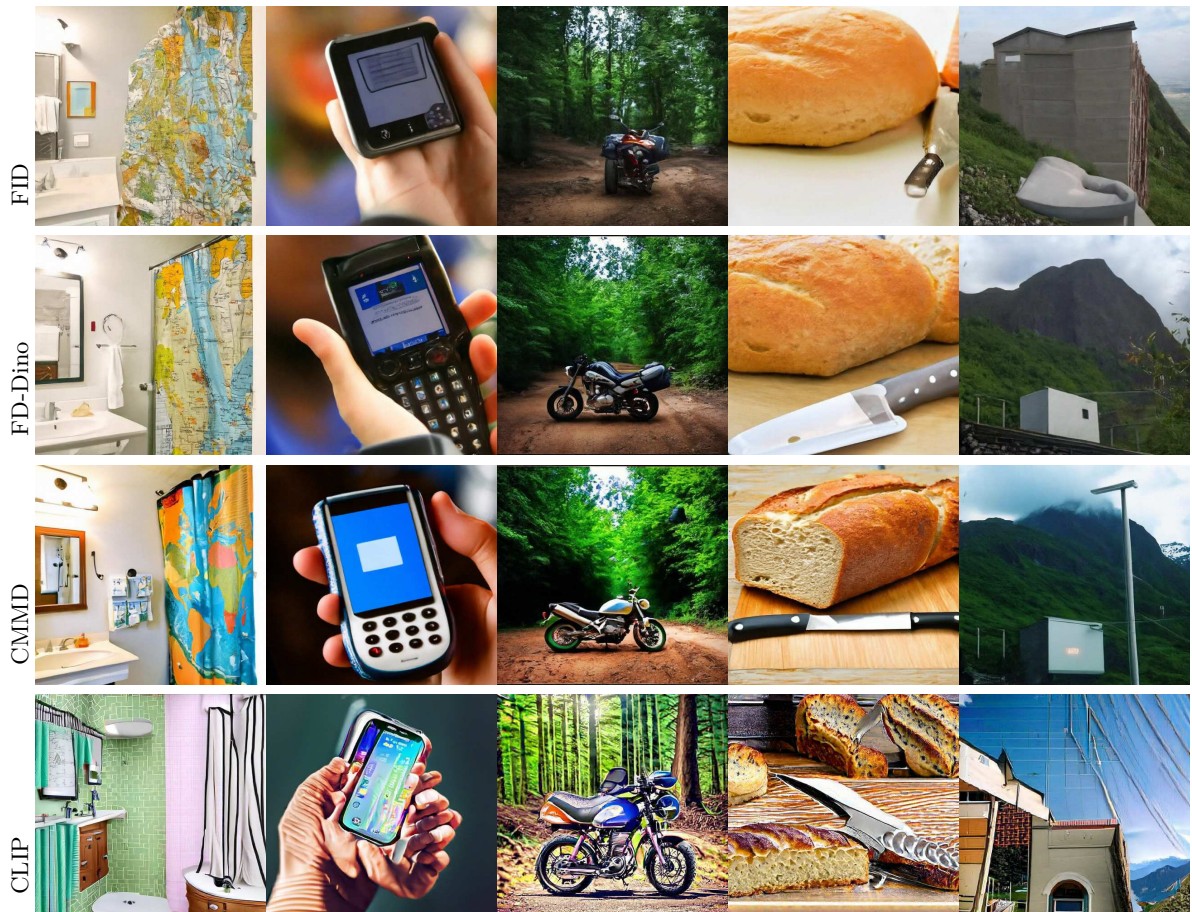

Figure 8: On the use of different metric spaces for calibrating guidance. First three rows: guidance values taken for minimizing respectively FID, FD-Dino and CMMD. The use of robust features is correlated with better shape and composition of images. Last row illustrates the side-effects of further increasing guidance and CLIP-score. Prompts from MSCOCO caption dataset: *'A bathroom with a sink and shower curtain with a map print.'*; *'A person holds a flip phone displaying the screen.'*; *'A motorcycle is parked on a dirt road in a forest.'*; *'A stainless shiny serrated knife sits in front of a sliced loaf.'*; *'A restroom hanging off the side of a building over a mountain.'*

## 6   A full diffusion pipeline: Vermeer

Vermeer is an 8B parameter model grown from 256 to 1024 pixel resolution. The UViT architecture is similar to our UViT-Huge model (Table 2), except that its bottom layers operate at a grid of $32x32$ and with 32 transformer blocks in total. We found that allocating transformer blocks at 32x scale improves details (like small faces). For Vermeer's text encoding, in addition to T5-XXL (Raffel et al., 2020a) and Clip (Radford et al., 2021b) embeddings previously mentioned, we also include a ByT5 (Xue et al., 2022b) encoder with 256 sequence length, resulting in a final embedding with sequence length of 461.

The baseline version (*Vermeer raw model*) is trained with 2k batch size at 256 resolution for 2M iterations, and grown to 1k resolution and finetuned for an additional 1M steps. As illustrated in Figure 1, it supports 3 aspect ratios, i.e., $1024 \times 1024$, $768 \times 1376$, and $1376 \times 768$ thought aspect ratio bucketing (Anlatan, 2022). Once the *raw model* is trained, we apply the following extra steps to improve the aesthetics of the generated images:

- Style finetuning. We train an image classifier based on images that conform to aesthetic and compositional attributes like those described in (Dai et al., 2023), and use it to select 3k images from our

training data as a fine-tuning set. We then fine-tune for 8K steps with a mixture of the original data and the aesthetic subset. We condition the model on the aesthetic subset by adding a token to the text prompt. We found that finetuning the pixel model with a mixture of pretraining and finetuning data is needed to avoid catastrophic forgetting and to avoid the introduction of additional artifacts.

- Distillation. The vanilla Vermeer model adopts 256-step sampling process, making it computationally expensive for real-world use. We employed the multistep consistency model (MCM) (Heek et al., 2024) to distill style-tuned Vermeer to 16 steps, achieving a substantial 16x speedup while maintaining high visual quality.

### 6.1 Vermeer results

We ablated four steps of Vermeer's development: (i) its raw model resulting from training on a large dataset; (ii) the result of applying prompt engineering at inference to the same model, adding words to improve aesthetic image quality, but with no further training; (iii) the final model, after style finetuning on a curated subset of 3k aesthetically pleasing images; and finally, (iv) its distilled, fast inference variation. Table 7 reports key performance metrics for all four variants, along with Stable Diffusion XL v1.0 (SDXL) (Podell et al., 2024). One can see that the raw model minimizes image distribution metrics that use state of the art feature space, i.e., FD-Dino and CMMD, while CLIP-score suggests a minor drop compared to SDXL. These metrics also highlight a significant shift away from the distribution of MSCOCO-captions (Chen et al., 2015), after augmenting the prompts (*+prompt engineering*) that is further increased when combined with the finetuning of the model for aesthetics pleasing image(*+style finetuning*).

The MSCOCO-captions dataset comprises reference image-caption pairs covering a diverse set of object categories and scenes. Thus, it offers an interesting distribution for measuring image quality and text alignment due to the complexity and diversity of the compositions. At the same time, its use for visual quality preference assessment is spurious as its images were not curated with human aesthetics preferences. On the contrary, many of the images have relatively poor aesthetic appeal. Thus, aiming to improve image aesthetics and composition, during Vermeer's prompt engineering and style tuning phases we intentionally

| model | | $\text{FID}_{30k} \downarrow$ | $\text{FD-Dino}_{30k} \downarrow$ | $\text{CMMD}_{30k} \downarrow$ | $\text{CLIP}_{score} \uparrow$ |
|---|---|---|---|---|---|
| $\text{SDXL}_{v1.0}$ | | **13.19** | 185.57 | 0.898 | **0.279** |
| Vermeer | *raw model* | 16.26 | **185.25** | **0.631** | 0.270 |
| | *+prompt engineering* | 17.33 | 216.01 | 0.867 | 0.269 |
| | *+style tuning* | 24.51 | 336.25 | 1.167 | 0.262 |
| | *distilled* | 25.97 | 347.19 | 0.885 | 0.261 |

Table 7: Image distribution metrics evaluated on 30k samples of MS-COCO. The raw Vermeer model minimizes distribution metrics that adopt feature spaces from SOTA models (FD-Dino uses Dino-v2 while CMMD adopts Clip features), while tuning it to produce aesthetically pleasing images intentionally diverges from MSCOCO distribution.

| model | | Entities | Relations | DSG↑ Attributes | Global | DSG |
|---|---|---|---|---|---|---|
| SD2.1 | | 75.44 | 53.06 | 69.66 | 68.49 | 71.23 |
| Muse | | 77.65 | 60.64 | 75.61 | 67.18 | 73.09 |
| Imagen Cascade | | 79.94 | 62.73 | 75.73 | 69.34 | 75.93 |
| $\text{SDXL}_{v1.0}$ | | 88.04 | 73.00 | 78.48 | 75.19 | 81.47 |
| Vermeer | *raw model* | 86.92 | 76.36 | 76.48 | 68.49 | 80.77 |
| | *+promp eng* | 87.94 | 74.92 | 76.31 | 67.41 | 80.99 |
| | *+style tunning* | 88.04 | 74.21 | 77.38 | 69.57 | 81.16 |
| | *+distillation.* | 84.71 | 69.23 | 72.68 | 65.49 | 76.88 |

Table 8: Vermeer. Broad and fine-grained results

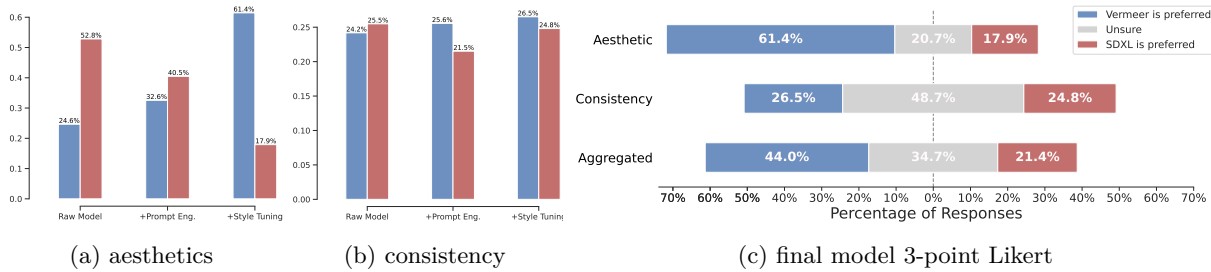

(a) aesthetics      (b) consistency      (c) final model 3-point Likert

Figure 9: **Human evaluation results**: Likert plot across 495 prompts, two tasks with 13 users each. Vermeer aesthetic is preferred during 61.4% of all comparisons, while its image-text consistency is marginally preferred. Aggregating the 1k annotations, Veermer is preferred during 44.0% of all comparisons, against 21.0% from SDXL. Prompt engineering and style tuning aligned with human preference for visual aesthetics. Side by side qualitative comparisons in Figure 12 and Figure 13.

move the distribution of images generated by Vermeer away from MSCOCO-caption distribution. To validate this we rely on human evaluation (in the next section).

The effect of the changes on the raw model with the CLIP-score and on semantic metrics on the other hand is minimal, aligned with our observation that the consistency of the model is not much affected by these two procedures. Semantic VqVa results are presented on Table 8. The references to Imagen (Saharia et al., 2022b) and Muse (Chang et al., 2023) models in this table are versions trained on internal data sources thus of similar resources and training pipelines than Vermeer. It shows that Vermeer presents competitive performance with SDXL, and surpassing the other models, including auto-regressive and cascade models.

Finally, we also develop a distilled version of our model, in order to offer an alternative version with faster inference time that similar to the other models presented in this paper operates as a single, non-cascade end-to-end model at inference time. Figure 1 illustrates Vermeer outputs and additional qualitative results including a comparison of samples from the full and distilled versions is presented in Appendix F.

## 6.2 Human evaluation

Assessing the performance of text-to-image models, ideally, depends on human evaluation, as this complex cognitive process necessitates a profound understanding of text and image relationships. Prior research has demonstrated that many recent works rely exclusively on automated metrics, such as the Fréchet Inception Distance (FID). However, it has been observed that the current automated measures are not fully consistent with human perception in assessing the quality of text-to-image samples (Otani et al., 2023). Thus, to objectively access the quality of images generated by Vermeer, we conduct a side-by-side human evaluation comparing our model with SDXL (Podell et al., 2024).

**Setup**. In this human evaluation, we ask annotators to evaluate generated images by Vermeer and SDXL based on the same prompt. For this, we collected 495 prompts [2] covering a range of skills: 160 are from TIFA v1.0 targeting measuring the faithfulness of a generated image to its text input covering 12 categories (object, attributes, counting, etc.)(Hu et al., 2023); 200 are sampled from the 1600 Parti Prompts (Yu et al., 2022), selecting for both complexity and diversity of challenges; and 150 others are created fresh for, or are sourced from, more recent prompting strategies targeting challenging cases.

We create two tasks in which we instruct annotators to consider either image quality (aesthetics) or fit to the prompt (consistency), and indicate their preferences using 3-point Liker scale: *Vermeer is preferred, Unsure*, and *SDXL is preferred* (the model names are anonymized). The neural response includes cases that both images are equally good and bad. In the annotation UI, the annotators are shown a prompt along with two images that are randomly shuffled. We collected 13 human ratings per prompt for both aesthetics and consistency (26 ratings per image).

---

[2]We first sampled 510 prompts, and 495 of them were usable after filtering incomplete samples.

**Results**. Prompt engineering and style tuning are confirmed to have a positive effect on human aesthetics preference (Figure 9, left), and small impact on text consistency (Figure 9, middle). They confirm our conjecture that the decrease on Vermeer's performance based on metrics grounded on the appearance of MSCOCO-caption dataset induced by these two steps are in alignment with the ultimate goal of human preference (Table 7).

Figure 9 (right) plots the Likert scale for our final model in each task (aesthetics or consistency) as well as the aggregated responses (shown in in the bottom bar). Overall, annotators prefer Vermeer 44% of the time, while they select SDXL 21.4% of the time, with relatively fewer *Neutral* responses (34.7%). Vermeer is clearly preferred for its aesthetics, with a win rate of 61.4%, while the gap in consistency between the two models is small, with a difference in the win rate of just 1.7%. Krippendorff's $\alpha$ for aesthetics and consistency are 0.27 and 0.41, respectively, indicating moderate agreement among annotators.

## 7 Conclusion

We propose a novel recipe for training non-cascaded large scale pixel-space text-to-image diffusion models. It benefits from splitting their training in two phases representing different tasks: learning image-text condition alignment and learning to generate images at high-resolution.

We identified the model *core components* as those responsible for the first task and propose a proxy architecture (Shallow-UViT) to supports its pretraining. The second task is learned with a *greedy growing* algorithm that stacks encoder-decoder layers of the final architecture on top of the pretrained *core components*. When learning the second task, our training recipe preserves the *core components* representation from the noise introduced by the grown layers and their random initialized weights.

Existing non-cascaded models training recipes struggle with scale, if not supported with large batch size and further regularization like dropout and multi-scale loss. Our approach is able to train models up to 8B parameters with small batch size (256) and no further regularization, by pretraining the *core components* and preserving it during the second training phase targeting high-resolution generation.

Compared with training from scratch and finetuning, the greedy growing procedure is more stable, and improves performance on a set of different metrics. Qualitative analysis shows that while keeping the *core components* representation stable it helps to preserve objects shape and overall structure, improving the definition of body parts. Our method allows use of data at different resolutions; the first phase benefits from the larger corpora with minimal requirements on image resolution, while the second phase learn to produce sharp images from the set filtered by the target resolution while reusing the representation learned from the larger set. We also explore models with increasing size, and show the benefits from scaling under different aspects and metrics.

In practice, the non-cascaded solution removes the out-of-distribution shift existent between training and deploying super-resolution phases. Based on that, we present Vermeer, an 8B parameter *Pixel based Text-to-Image Diffusion Model* that produces high-resolution high-quality images using a single non-cascaded model. By training it on a larger dataset, and incorporating a final style tuning phase, Vermeer is able to surpass SDXL v1.0 in human preference study.

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
