# A    Teaser image prompts

Next, we list the prompts used for generating images at Figure 1 using Vermeer. Their corresponding location is shown in Table 9).

1. the word 'START' written in chalk on a sidewalk
2. a basketball to the left of two soccer balls on a gravel driveway
3. An Egyptian tablet shows an automobile.
4. Macro photography of rose, centered, mini, dark tones, drops of water, cannon
5. photo of a woman's face floating in the water with her eyes closed, you can only see top part of her face above water, reflections, abstract conceptual, realistic reflection, pale sky, scientific photo, high quality fantasy stock photo
6. cyberpunk starship troopers cinematic 4d
7. 3-d Letter "O" made from orange fruit, studio shot, pastel orange background, centered
8. 3-d Letter "W" made from transparent water, studio shot, pastel light blue background, centered
9. 3-d Letter "T" made from tiger fur, studio shot, pastel orange background, centered.
10. Many people carry sacks along a trail through a bright field with long grass and flowers and muted tones. Two small cottages. Dark row of trees. Green hills, blue sky, clouds. Pastoral landscape. Ein plein air. Vibrant, saturation, free brush strokes. Impressionism. Oil on canvas by Auguste Renoir.
11. a photograph of a blue porsche 356 coming around a bend in the road
12. photography of a cat sitting at a sushi restaurant, wearing a blue coat and taking sushi from the boat. Neon bright light, high contrast, low vibrance
13. turtle with German Shepherd dog's head growing from it, DSLR
14. A futuristic street train a rainy street at night in an old European city. Painting by David Friedrich, Claude Monet and John Tenniel.
15. building behind train
16. Realistic photograph of a cute otter zebra mouse in a field at sunset, tall grass, macro 35mm film
17. A 1920's race car with number 7 parked near a fountain in a modern city. Painting by David Friedrich, Claude Monet and John Tenniel.
18. The clock on the bricked building is green. The numbers are in roman numerals. The details have gold accents. The bricked building has a window beside the clock.
19. duck with rabbit's head growing from it, DSLR
20. cauliflower with sheep's head growing from it, DSLR
21. Silver 1963 Ferrari 250 GTO in profile racing along a beach front road. Bokeh, high-quality 4k photograph.
22. a photograph of a knight in shining armor holding a basketball

|   |   |   |   |   |
|---|---|---|---|---|
| 1 | 3 | 4 | 6 | 7 |
| | | 5 | | 8 |
| 2 | | | | 9 |
| 10 | 11 | 12 | 14 | 16 |
| | | 13 | 15 | |
| 17 | 18 | 19 | | 22 |
| | | 20 | 21 | |

Table 9:  Map of prompts used to generate Vermeer results illustrated in Figure 1

## B  Shallow-UViT: Vqva detailed categories

This appendix complements the results on DSG's broad categories presented in the main text by providing their corresponding fine grain results. That is, Table 4 results are complemented with the fine grain detailed results in Table 10, Table 6 with Table 11 and Table 8 with Table 12.

| Model | | VqVa Question types | | | | | | | | | | | | | DSG |
|---|---|---|---|---|---|---|---|---|---|---|---|---|---|---|---|
| | whole | part | spatial | shape | color | state | type | count | text rendering | texture | global | material | scale | size | |
| # prompts | 2851 | 517 | 1477 | 84 | 432 | 740 | 173 | 196 | 116 | 40 | 649 | 92 | 25 | 11 | |
| Shallow-UViT Base | 0.567 | 0.412 | 0.333 | 0.417 | 0.550 | 0.409 | 0.402 | 0.523 | 0.487 | 0.450 | 0.400 | 0.272 | 0.500 | 0.318 | 48.08 |
| Shallow-UViT Large | 0.626 | 0.451 | 0.395 | 0.446 | 0.579 | 0.478 | 0.454 | 0.554 | 0.552 | 0.475 | 0.437 | 0.353 | 0.600 | 0.364 | 52.54 |
| Shallow-UViT Huge | 0.706 | 0.617 | 0.488 | 0.548 | 0.624 | 0.530 | 0.509 | 0.587 | 0.552 | 0.562 | 0.433 | 0.424 | 0.740 | 0.409 | 60.25 |
| Shallow-UViT XHuge | 0.724 | 0.617 | 0.518 | 0.577 | 0.646 | 0.568 | 0.540 | 0.582 | 0.591 | 0.562 | 0.441 | 0.424 | 0.820 | 0.636 | 61.91 |

Table 10: Shallow-UViT scaling: DSG fine-grained semantic categories. DSG: average score accross DS1K images.

| Model | | VqVa Question types | | | | | | | | | | | | | DSG |
|---|---|---|---|---|---|---|---|---|---|---|---|---|---|---|---|
| | | whole | part | spatial | shape | color | state | type | count | text rendering | texture | global | material | scale | size |
| | # prompts | 2851 | 517 | 1477 | 84 | 432 | 740 | 173 | 196 | 116 | 40 | 649 | 92 | 25 | 11 |
| UVIT-Base | scratch | 0.743 | 0.671 | 0.540 | 0.655 | 0.635 | 0.639 | 0.552 | 0.564 | 0.690 | 0.575 | 0.555 | 0.489 | 0.780 | 0.545 | 64.83 |
| | finetuning | 0.723 | 0.587 | 0.500 | 0.560 | 0.597 | 0.596 | 0.590 | 0.628 | 0.625 | 0.525 | 0.532 | 0.478 | 0.700 | 0.500 | 62.75 |
| | frozen | 0.702 | 0.666 | 0.495 | 0.560 | 0.544 | 0.647 | 0.566 | 0.584 | 0.591 | 0.613 | 0.534 | 0.332 | 0.640 | 0.409 | 61.16 |
| | freeze-unfreeze | 0.748 | 0.657 | 0.536 | 0.649 | 0.650 | 0.645 | 0.587 | 0.640 | 0.694 | 0.625 | 0.569 | 0.418 | 0.620 | 0.455 | 66.13 |
| UVIT-Large | scratch | 0.750 | 0.642 | 0.521 | 0.631 | 0.642 | 0.640 | 0.618 | 0.643 | 0.547 | 0.562 | 0.580 | 0.511 | 0.640 | 0.591 | 66.02 |
| | finetuning | 0.761 | 0.688 | 0.542 | 0.601 | 0.681 | 0.678 | 0.604 | 0.666 | 0.638 | 0.650 | 0.579 | 0.473 | 0.780 | 0.636 | 67.39 |
| | frozen | 0.800 | 0.730 | 0.616 | 0.643 | 0.738 | 0.684 | 0.618 | 0.648 | 0.728 | 0.700 | 0.614 | 0.418 | 0.760 | 0.500 | 72.13 |
| | freeze-unfreeze | 0.761 | 0.668 | 0.556 | 0.571 | 0.646 | 0.655 | 0.665 | 0.717 | 0.616 | 0.625 | 0.588 | 0.473 | 0.840 | 0.545 | 67.79 |
| UVIT-Huge | scratch | 0.758 | 0.662 | 0.551 | 0.595 | 0.634 | 0.645 | 0.627 | 0.607 | 0.698 | 0.600 | 0.586 | 0.505 | 0.840 | 0.591 | 66.90 |
| | finetuning | 0.773 | 0.775 | 0.565 | 0.548 | 0.684 | 0.716 | 0.705 | 0.648 | 0.659 | 0.650 | 0.626 | 0.484 | 0.640 | 0.591 | 69.67 |
| | frozen | 0.827 | 0.814 | 0.648 | 0.649 | 0.749 | 0.722 | 0.685 | 0.635 | 0.797 | 0.688 | 0.619 | 0.500 | 0.820 | 0.500 | 75.15 |
| | freeze-unfreeze | 0.798 | 0.748 | 0.582 | 0.583 | 0.675 | 0.684 | 0.653 | 0.666 | 0.711 | 0.575 | 0.609 | 0.467 | 0.780 | 0.500 | 71.50 |
| UVIT-XHuge | freeze | **0.840** | **0.821** | **0.668** | **0.637** | **0.744** | 0.720 | **0.720** | **0.671** | **0.668** | **0.688** | 0.629 | 0.473 | **0.860** | 0.455 | **75.70** |
| | freeze-unfreeze | 0.817 | 0.783 | 0.607 | 0.631 | 0.722 | 0.705 | 0.679 | 0.681 | 0.681 | 0.675 | 0.602 | 0.533 | 0.880 | 0.727 | 73.53 |
| SD2.1 | | 0.760 | 0.730 | 0.530 | 0.679 | 0.707 | 0.729 | 0.665 | 0.571 | 0.655 | 0.637 | 0.685 | 0.495 | 0.780 | 0.455 | 71.23 |

Table 11: End-to-end models: DSG fine-grained semantic categories. DSG: average score accross DS1K images.

| Model | | VqVa Question types | | | | | | | | | | | | | DSG |
|---|---|---|---|---|---|---|---|---|---|---|---|---|---|---|---|
| | | whole | part | spatial | shape | color | state | type | count | text rendering | texture | global | material | scale | size |
| Muse | | 0.780 | 0.761 | 0.605 | 0.714 | 0.814 | 0.766 | 0.668 | 0.610 | 0.651 | 0.838 | 0.672 | 0.647 | 0.780 | 0.773 | 73.09 |
| SD2.1 | | 0.760 | 0.730 | 0.530 | 0.679 | 0.707 | 0.729 | 0.665 | 0.571 | 0.655 | 0.637 | 0.685 | 0.495 | 0.780 | 0.455 | 71.23 |
| Imagen Cascade | | 0.799 | 0.806 | 0.626 | 0.714 | 0.806 | 0.772 | 0.723 | 0.673 | 0.750 | 0.738 | 0.693 | 0.641 | 0.820 | 0.636 | 75.93 |
| Imagen Vermeer | raw | 0.884 | 0.787 | 0.765 | 0.690 | 0.810 | 0.737 | 0.798 | 0.689 | 0.789 | 0.787 | 0.685 | 0.701 | 0.940 | 0.773 | 80.77 |
| | + prompt eng. | 0.892 | 0.810 | 0.751 | 0.750 | 0.840 | 0.732 | 0.809 | 0.679 | 0.784 | 0.825 | 0.674 | 0.625 | 0.880 | 0.591 | 80.99 |
| | + style tuning | 0.889 | 0.833 | 0.744 | 0.696 | 0.836 | 0.747 | 0.818 | 0.714 | 0.716 | 0.838 | 0.696 | 0.707 | 0.840 | 0.591 | 81.16 |

Table 12: Vermeer: DSG fine-grained semantic categories .

## C  On validating the representation quality improvements from scale by counting

Given the importance of counting and other basic numerical skills in biological intelligence (Nieder & Dehaene, 2009), we expect that competitively performing T2I show similar behaviour when evaluated on such skills. Counting requires manipulation of abstract concepts (numbers) and evaluating this ability provides an objective measure of a well-defined skill. As such it is easier to evaluate and interpret the performance of the model on the counting task, in contrast to some other image characteristics such as aesthetics that might depend on an individual's preferences.

To evaluate models' ability to correctly generate an image with an exact number of objects, we use 59 prompts in the *att/count* category of the Gecko benchmark (Wiles et al., 2024). The Gecko benchmark aims to comprehensively and systematically probe T2I model alignment along different skills such as numerical and spatial reasoning, text rendering, depicting of colors and shapes, and many others.

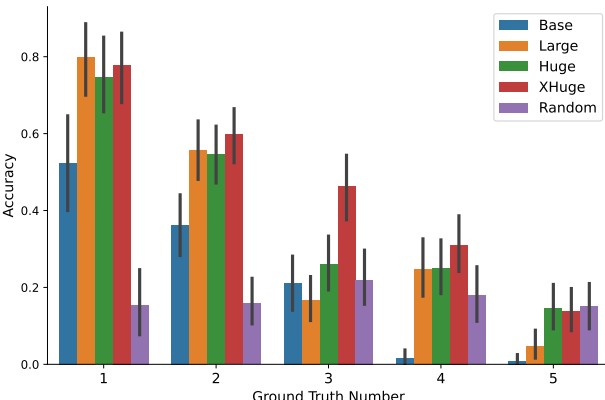

Figure 10: Breakdown of accuracy per number in the original prompt used to generate the image.

Specifically, our analyses include 48 *simple modifier* prompts and 11 *additive* prompts with numbers between 1 and 5. *Simple modifier* prompts are of form "*num noun*" (eg. "1 cat"), where *num* is a number represented by a single digit (ie. 1, 2, 3) or a numeral (ie. "one", "two" or "three") and the noun is a word from a common natural semantic categories such as foods, animals and everyday objects. *Additive* prompts are compositions of individual *simple modifier* prompts as they combine two nouns and two numbers, such as "1 cat and 3 dogs". By using such systematically curated prompts, we are implicitly testing whether models can count, as the ability to correctly generate a number of objects depends on the ability to keep track of objects that were already generated.

To evaluate the correctness of T2I generation of numbers, we recruit human raters through a crowd-sourcing platform to provide the count of objects in every generated image. The study design, including remuneration for the work were reviewed and approved by our institution's independent ethical review committee. We collect 5 annotations per generated image by asking "How many X are there in the image?" where X is the object mentioned in the original prompt used to generate that image. We generate three images for each prompt and each model using different seeds.

Figure 10 shows the breakdown of accuracy per model type as well as per the ground truth number. The ground truth number is the number in the original prompt used to generate the image. The accuracy is the average number of annotations that match the ground truth label for a question and a given model. We observe that all models (with the exception of Base) perform comparably well on generating images with only one object, but this deteriorates with higher number, and only XHuge is able to correctly generate number 3 above the chance level. While exact number generation appears to improve with scale, it is unclear whether this pattern saturates for higher numbers.

## D    Qualitative comparison of finetuning and frozen e2e models

Our qualitative comparison between finetuning and frozen *core components* is based on 50 prompts covering different animal species. They are chosen for covering a diverse set of shapes, textures and structures. Figure 11 present a side by side comparison at 50k training steps using the UVIT-Huge model. Structural elements like legs, wings and trunks are better formed when freezing the pretraing *core components* representation. The images were produced with the following list of prompts.

1. "A majestic lion with a flowing mane, basking in the golden African sunset."
2. "A playful dolphin leaping out of the water, glistening with droplets."
3. "A wise old owl perched on a moonlit branch, gazing with piercing yellow eyes."
4. "A colorful macaw soaring through a lush, vibrant rainforest."
5. "A mischievous raccoon rummaging through a trash can in a suburban backyard."

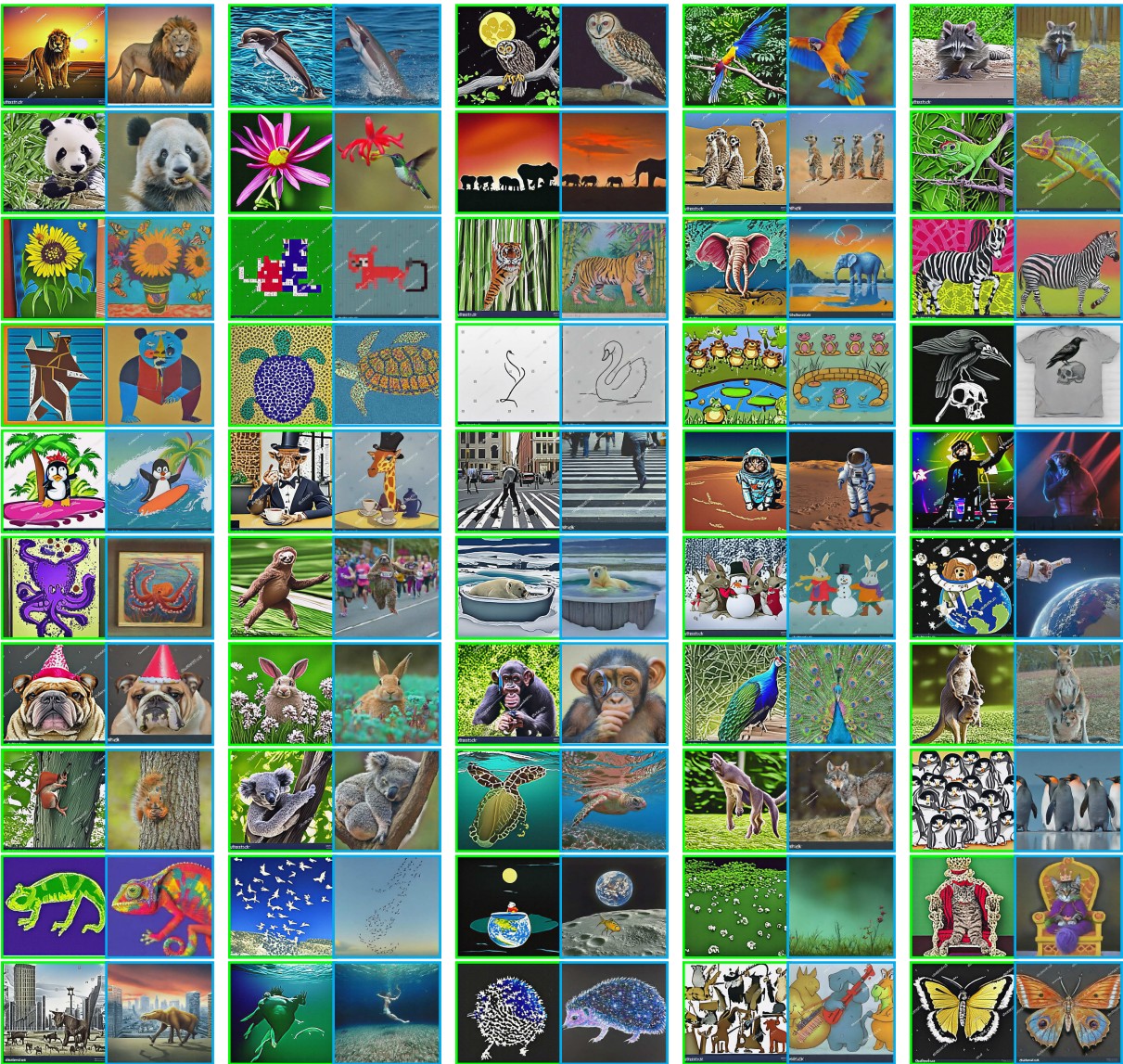

Figure 11: On the reuse of the core layers: qualitative results. Finetune (green bounding boxes) × Frozen results (blue bounding boxes) at $50k$ steps. Images at $512 \times 512 pixels$. Models trained with CC12M. Freezing the representation induces objects with better global and parts structure, from the very early steps of training.

6. "A close-up portrait of a fluffy panda munching on bamboo."

7. "A graceful hummingbird hovering near a bright pink flower."

8. "A herd of elephants silhouetted against a fiery orange sky."

9. "A group of meerkats standing alert in the desert, looking out for danger."

10. "A photorealistic image of a chameleon blending seamlessly with its surroundings."

11. "A Van Gogh-inspired painting of sunflowers with butterflies flitting around them."

12. "A pixel art rendition of a pixelated cat chasing a pixelated mouse."

13. "A watercolor painting of a majestic tiger stalking through a bamboo forest."

14. "A surreal landscape with a melting elephant in the style of Salvador Dalí."

15. "A vibrant pop art image of a zebra with bold stripes and contrasting colors."

16. "A cubist artwork depicting a fragmented and reassembled bear."

17. "A pointillist painting of a turtle, created with tiny dots of color."

18. "A minimalist line drawing of a graceful swan."

19. "A whimsical cartoon illustration of a group of singing frogs in a pond."

20. "A dark and gothic illustration of a raven perched on a skull."

21. "A penguin riding a surfboard on a giant tropical wave."

22. "A giraffe wearing a top hat and monocle, enjoying a cup of tea in a fancy cafe."

23. "A zebra crossing a busy city street at a crosswalk."

24. "A cat wearing a space suit, exploring the surface of Mars."

25. "A monkey DJ mixing beats at a neon-lit dance club."

26. "An octopus painting a self-portrait with its many arms."

27. "A sloth running a marathon, surprisingly outrunning all competitors."

28. "A polar bear relaxing in a hot tub in the middle of the Arctic."

29. "A group of rabbits building a snowman in a winter wonderland."

30. "A dog astronaut floating in space, gazing at the Earth."

31. "A grumpy bulldog wearing a birthday hat and refusing to smile."

32. "A joyful rabbit hopping through a field of wildflowers."

33. "A curious chimpanzee looking intently through a magnifying glass."

34. "A proud peacock displaying its magnificent tail feathers."

35. "A loving mother kangaroo carrying her joey in her pouch."

36. "A mischievous squirrel hiding nuts in a tree trunk."

37. "A sleepy koala clinging to a tree branch, taking a nap."

38. "A determined sea turtle swimming against the ocean current."

39. "A playful wolf pup chasing its own tail."

40. "A group of penguins waddling together in a comical huddle."

41. "A chameleon painted with the vibrant colors of a bustling city skyline." (Imagine a chameleon camouflaged with neon signs and skyscraper patterns.)

42. "A flock of birds forming the shape of a musical note in flight." (Visualize a dynamic dance of birds creating a melody in the sky.)

43. "A fishbowl on the moon, with an astronaut goldfish gazing at Earth." (A whimsical and thought-provoking perspective shift.)

44. "A microscopic landscape teeming with life, where insects are giants and blades of grass are towering trees."

45. "A cat wearing a crown and royal robe, sitting regally on a throne made of yarn balls." (A playful portrait with a touch of humor.) **

46. "A photorealistic image of extinct animals roaming in a modern city landscape." ** (Blend the past and present for a surreal scene.)

47. "An underwater ballet performed by graceful sea creatures." (Capture the beauty and movement of marine life in an artistic way.)

48. "A hedgehog painted as a starry night sky, with its spines representing twinkling stars." (A dreamy fusion of nature and the cosmos.)

49. "Animals playing musical instruments together in a harmonious orchestra." (Imagine the symphony created by a unique animal band.)

50. "A close-up portrait of a butterfly, revealing the intricate patterns and textures on its wings in exquisite detail." (Appreciate the delicate beauty of nature.)

# E    Human evaluation

This appendix complements the human evaluation quantitative results presented in subsection 6.2, with some illustrations of the examples compared. Figure 12 and Figure 13 illustrate winning, losing, and matching cases for respectively the visual aesthetic and text consistency tasks. During the ranking, the image order was shuffled, but kept in fixed ordered in these illustrations for readability only. Pairs were selected to illustrate responses observed to reach 'Perfect agreement' (all raters agreed with the winning class), 'Strong agreement' (12-10 of the 13 raters agreed), 'Majority agreement' (9-7 of the 13 raters agreed) and 'Weak agreement' (6-5 of the 13 raters agreed on the winning class). They illustrate the model's and the preference agreement variability across different prompts. Thus, these qualitative examples further sustain our choice for covering a large number and diverse set of prompts on our human evaluation settings. While raters clearly agree when one of the images contains the presence of an artifact (see Figure 12 SDXL's Treehouse and panda bear with multiple glasses) or cases in which the models clearly have not followed the text description (In Figure 13 first row Vermeer image is missing the microphone, while SDXL raccoon lizard shows a regular raccoon), some prompts lead to generations that can be argued as equally good (as in the forest path in Figure 12) or equally bad (as in "two laptops a mouse cords wires and a monitor").

We favored the coverage of a large number of challenging prompts following the observation that the different models' performance vary across different query modes. Overall, evaluating human preferences in generative models remains a complex and unresolved issue. To our knowledge, there is no widely accepted protocol for conducting such evaluations. The development of such protocol would benefit the community but is out of the scope or this report.

# F    Vermeer distillation: qualitative results

Figure 14 presents additional qualitative results produced using the Vermeer model and its distilled version.

The images were produced with the following list of prompts.

1. "Ruined circular stone tower on a cliff next to the ocean. Shepherd and sheep on green hillock. Sunrise, big puffy clouds. Naturalistic landscape. Romanticism. Hudson River School. Oil on canvas by Thomas Cole."
2. "Photo of a cute raccoon lizard at sunset, 35mm"
3. "Wallpaper of minimal origami corgi made of multi colored paper, abstract, clean, minimalist, 4K, 8K, soft colors, high definition."
4. "A cat lying a top on the desk on a laptop."
5. "A green stop sign on a pole."
6. "A grey motorcycle on dirt road next to a building."
7. "'Fall is here' written in autumn leaves floating on a lake."
8. "A cake topped with whole bulbs of garlic"
9. "A red plate topped with broccoli, meat and veggies."
10. "A photorealistic image of a chameleon blending seamlessly with its surroundings."
11. "A cat wearing a cowboy hat and sunglasses and standing in front of a rusty old white spaceship at sunrise. Pixar cute. Detailed anime illustration."
12. "A pizza with cherry toppings"

Perfect agreement

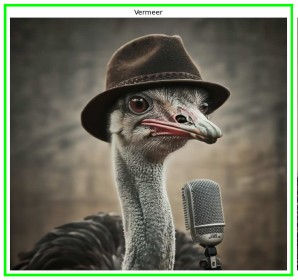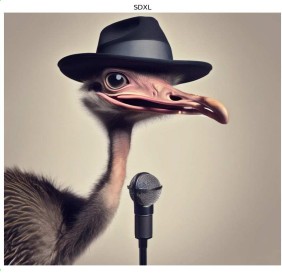 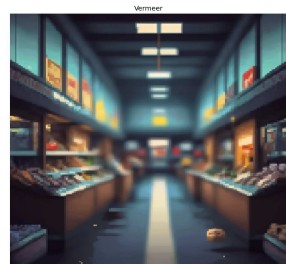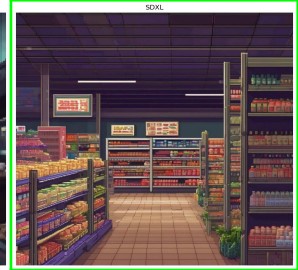

Parti: "a photograph of an ostrich wearing a fedora and singing soulfully into a microphone"

Parti: "pixel art scene, a quiet and empty supermarket at night, atmospheric, 16-bit"

Strong agreement

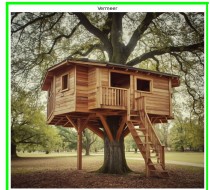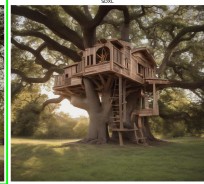 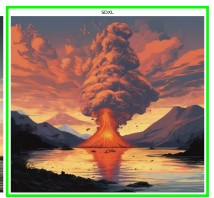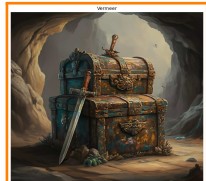 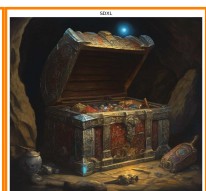

Tifa: "a wood treehouse in an oak tree"

Parti: "a volcano spewing fish into the sky"

Parti: "a painting of an ornate treasure chest with a broad sword propped up against it, glowing in a dark cave"

Majority agreement

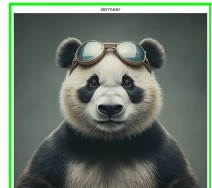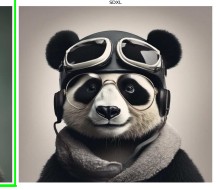 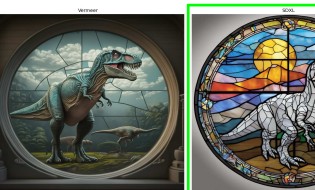 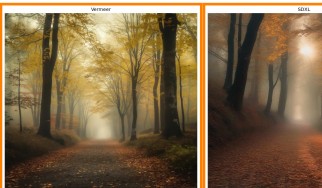

Tifa: "a panda bear with aviator glasses on its head

Tifa: "a stained glass window depicting a calm tyrannosaurus rex"

Parti: "misty autumn forest path, fallen leaves carpeting the ground, sunlight filtering through the trees at daybreak"

Weak agreement

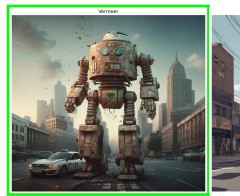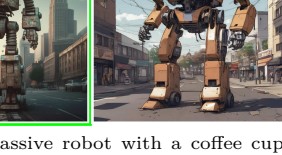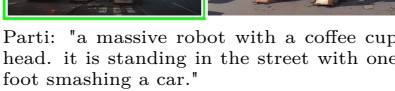 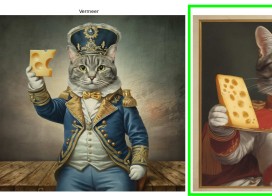 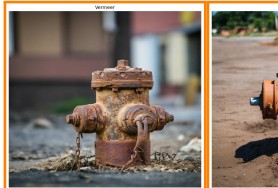

Parti: "a massive robot with a coffee cup head. it is standing in the street with one foot smashing a car."

Parti: "a propaganda poster depicting a cat dressed as french emperor napoleon holding a piece of cheese"

Tifa: "a rusty fire hydrant surrounded by dirt"

Figure 12: Human evaluation pairs, aesthetics preference qualitative results: Vermeer (left), SDXL (right). Green boxes indicate preference cases. Neutral preferences are indicated in orange. Agreement levels indicate the fraction of the 13 evaluators associated with the winning class among three options: "Vermeer is preferred", "SDXL is preferred", and the class indicating neutral preference: "Unsure". Perfect agreement: 100%, Strong agreement: $92 - 77\%$, Majority agreement: $70 - 54\%$, "Weak agreement": $47 - 38\%$. No image has perfect agreement on the neutral preference as neutral.

Perfect agreement

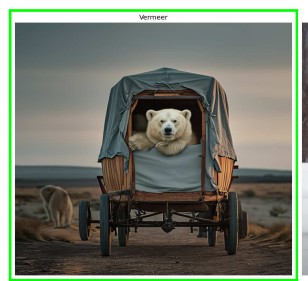 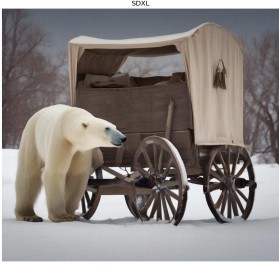

Parti: "a photo of the back of a covered wagon. a polar bear is sticking it's head out of the wagon."

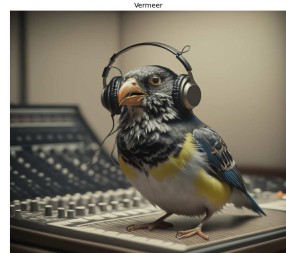 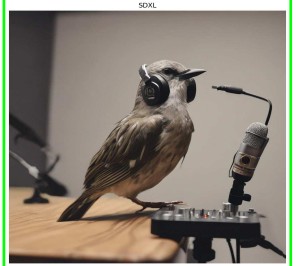

Parti: "a photograph of a bird wearing headphones and speaking into a microphone in a recording studio"

Strong agreement

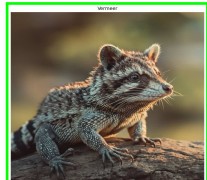 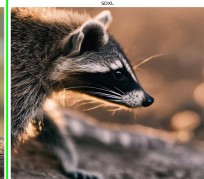

Parti: "photo of a cute raccoon lizard at sunset, 35mm"

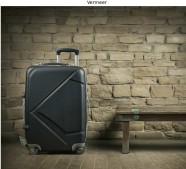 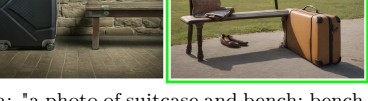

Tifa: "a photo of suitcase and bench; bench is left to suitcase"

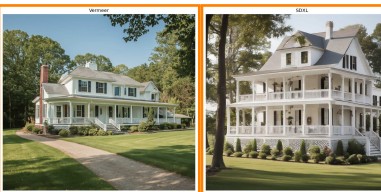

Tifa: "a white country home with a wraparound porch"

Majority agreement

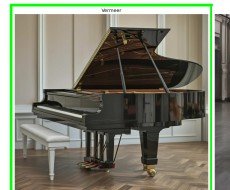 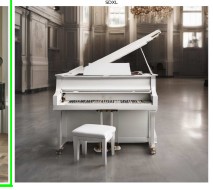

Tifa: "a grand piano with a white bench"

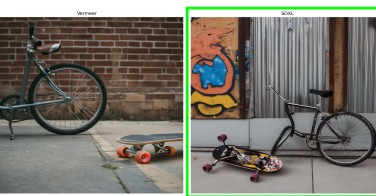

Tifa: "a photo of bike and skateboard; skateboard is left to bike"

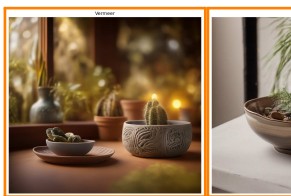

Parti: "commercial photography, a handcrafted ceramic bowl, earth tones, soft lighting, plants"

Weak agreement

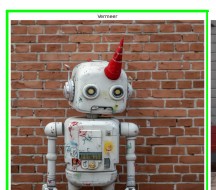 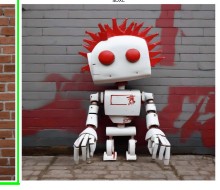

Parti: "a white robot with a red mohawk painted as graffiti on a red brick wall."

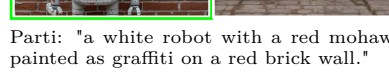 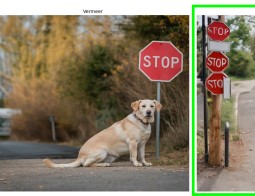

Tifa: "a photo of dog and stop sign; stop sign is left to dog"

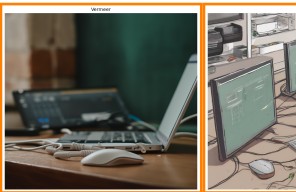

Tifa: "two laptops a mouse cords wires and a monitor"

Figure 13: Human evaluation pairs, consistency preference qualitative results: Vermeer (left), SDXL (right). Green boxes indicate preference cases. Neutral preferences are indicated in orange. Agreement levels indicate the fraction of the 13 evaluators associated with the winning class among three options: "Vermeer is preferred", "SDXL is preferred", and the class indicating neutral preference: "Unsure". Perfect agreement: 100%, Strong agreement: 92 − 77%, Majority agreement: 70 − 54%, "Weak agreement": 47 − 38%. No image has perfect agreement on the neutral preference as neutral.

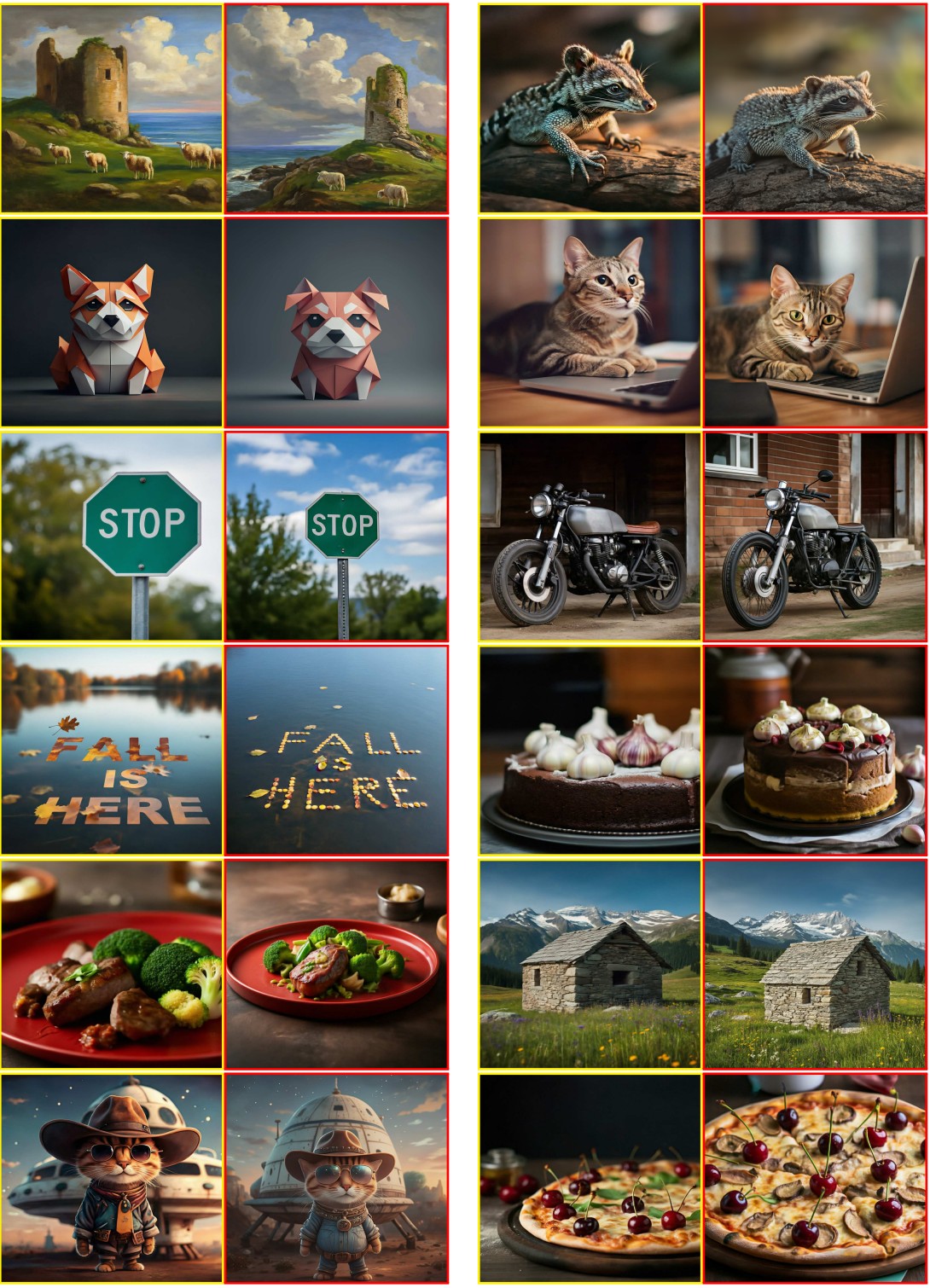

Figure 14: Qualitative comparison between style-tuned Vermeer using 256 steps (red bounding boxes) and its distilled MCM version using 16 steps (yellow bounding boxes). All images are directly generated at 1024x1024 pixels.