# OpenReview forum: "Greedy Growing Enables High-Resolution Pixel-Based Diffusion Models"
_TMLR — Accepted by TMLR_

### Review · Reviewer_Mr9y · 2024-06-16

**Summary Of Contributions:**

This paper deals with the challenge of training large-scale pixel-space text-to-image diffusion models (PSDM). The key idea is to decouple the alignment of image-text concepts from generating high-resolution images. Based on this insight, the authors propose a novel framework that enables pretraining PSDM’s core layers without requiring many high-res images. The authors further propose an algorithm to train the PSDM to generate high-res images. The resulting approach overcomes issues of existing methods and is able to achieve comparable performance with the current state-of-the-art models.

**Audience:**

Yes

**Broader Impact Concerns:**

I have no concerns about the ethical implications of the work.

**Claims And Evidence:**

Yes

**Requested Changes:**

See the weaknesses stated above.

Minor:

- Abstract: “…a remarkably simple greedy method for stable training of largescale, high-resolution models. without the needs for …”: remove the dot before “without”.

**Strengths And Weaknesses:**

Strengths

- The paper clearly stated the issue of existing methods, including the requirement for large batch size, demand for high-quality high-res data, and inapplicability for tasks using diffusion models for image priors.

- The paper is well-structured, with nice figures and tables.

- The paper provides comprehensive ablation studies on the model size.


Weaknesses

- The presentation of the methodology part is somewhat hard to follow. It could be improved with more use of formal notations and equations.

- The proposed Vermeer only beats SDXL v1.0 on human evaluation, which can be very subjective and random, especially considering there are only 13 ratings for each prompt. I also wonder why the authors do not show the generated images by Vermeer and SDXL on the same prompts.

- In the introduction section, the authors claim that cascaded diffusion models are not applicable to tasks where diffusion models serve as image priors. So, I wonder how the proposed model performs when serving as image priors for downstream tasks.

---

> ### Author Response · Authors · 2024-08-05
> **The latest update features a side-by-side comparison within the same prompt as requested.**
>
> Thank you for the encouraging review and highlight of our paper's main contributions. In the following we hope to address your concerns. We altered the paper to address your suggestions. If there is anything missing, please let us know.
>
> About the requested changes:
> We fixed the typo in the abstract. Thanks for pointing that out!
>
> About the listed weaknesses:
>
> Evaluating human preferences in generative models remains a complex and unresolved issue. We acknowledge that ensuring consistency among annotators is crucial for reliable results. To our knowledge, there is no widely accepted protocol for conducting such evaluations. We favored the coverage of a large number of challenging prompts following the observation that the different models’ performance vary across different query modes.  By comparison: here’s what others literature references have done, by the numbers, for human eval:
> - Imagen [Chitwan 2022] adopted two tasks with 25 raters per each of DrawBench’s 11 categories, covering 200 prompts and two tasks with a total of 10.000  ratings.
> - SDXL [Podell 2023] asked raters to compare a single task (text adherence), and collected a large number of evaluations (17,153), but covering 30 prompts only.
> - SD3-Turbo [Sauer 2024] adopted 4 raters over 128 prompts and asked about image quality and human alignment with a total of 1024 ratings.
> - Dalle 3 [Betker 2023] adopted 170 prompts and three tasks with a total of 6.120 ratings.
>
> Our evaluation encompassed a total of 13000 ratings concerning both aesthetics and prompt alignment covering 500 prompts curated as challenging for t2i models, and covering diverse categories  [Yu 2022, Hu 2023]. We highlight that  our evaluation included the most extensive collection of prompts compared to the mentioned models. Additionally, we utilized a comparable or, in most cases, a larger number of annotations, ensuring the robustness and reliability of our assessment.
>
> Following your suggestion, we've updated the paper and included images by Vermeer and SDXL on the same prompts. Figures 12 and 13 illustrate side-by-side examples of human evaluation results. These examples illustrate various prompts where raters either preferred our model, found no clear winner, or favored SDXL. Kindly review the new version for these additions.
>
> ___
>
> Concerning your final observation, exploring the potential of the proposed model as image priors presents an exciting avenue for future research, however, it falls beyond the scope of the current report.
>
> ________________________
>
> [Chitwan 2022] Saharia, Chitwan and Chan, William and Saxena, Saurabh and Li, Lala and Whang, Jay and Denton, Emily L and Ghasemipour, Kamyar and Gontijo Lopes, Raphael and Karagol Ayan, Burcu and Salimans, Tim and Ho, Jonathan and Fleet, David J and Norouzi, Mohammad. S. Koyejo and S. Mohamed and A. Agarwal and D. Belgrave and K. Cho and A. Oh. Photorealistic Text-to-Image Diffusion Models with Deep Language Understanding}, NeuRIPS 2022.
>
> [Podell 2023] Dustin Podell, Zion English, Kyle Lacey, Andreas Blattmann, Tim Dockhorn, Jonas Müller, Joe Penna, Robin Rombach. SDXL: Improving Latent Diffusion Models for High-Resolution Image Synthesis. arXiv 2023.
>
> [Sauer 2024] Axel Sauer, Frederic Boesel, Tim Dockhorn, Andreas Blattmann, Patrick Esser, Robin Rombach. Fast High-Resolution Image Synthesis with Latent Adversarial Diffusion Distillation. arXiv 2024.
>
> [Betker 2023]  James Betker, Gabriel Goh, Li Jing, Tim Brooks, Jianfeng Wang, Linjie Li, Long Ouyang, Juntang Zhuang, Joyce Lee, Yufei Guo, Wesam Manassra, Prafulla Dhariwal, Casey Chu, Yunxin Jiao, Aditya Ramesh. Improving Image Generation with Better Captions. Arxiv 2023.
>
> [Hu 2023] Yushi Hu, Benlin Liu, Jungo Kasai, Yizhong Wang, Mari Ostendorf, Ranjay Krishna, Noah A. Smith. TIFA: Text-to-Image Faithfulness Evaluation with Question Answering.  ICCV 2023
>
> [Yu 2022] Jiahui Yu, Yuanzhong Xu, Jing Yu Koh, Thang Luong, Gunjan Baid, Zirui Wang, Vijay Vasudevan, Alexander Ku, Yinfei Yang, Burcu Karagol Ayan, Ben Hutchinson, Wei Han, Zarana Parekh, Xin Li, Han Zhang, Jason Baldridge, Yonghui Wu. Scaling Autoregressive Models for Content-Rich Text-to-Image Generation. Transactions on Machine Learning Research (TMLR 2022).

---

### Review · Reviewer_BAm2 · 2024-06-20

**Summary Of Contributions:**

The authors introduce a method for training high-resolution, pixel-based image diffusion models.  The approach allows a single stage model to generate high-resolution images without a large high-resolution dataset or cascaded super-resolution components.  The generated images are preferred by human evaluators 2:1 over SDXL.  In comparison to cascaded models, this approach avoids the domain shift and propagation of irregular distortions in higher levels.  Their architecture, Shallow-UViT enables pretraining of the PSDM's core on text-image data as opposed to requiring a massive high resolution image dataset.  Their greedy algorithm enables the text-to-image model to be trained with small batch sizes.  The approach to frow the architecture produces better image distribution, quality, and text alignment.

**Audience:**

Yes

**Broader Impact Concerns:**

No concerns

**Claims And Evidence:**

Yes

**Requested Changes:**

Typo in intro in first sentence- period between "models. without"
Typo in "Full pipeline model: ... " reply --> rely
Backwards quotes throughout
Figure 3, Step 1.75M is not aligned in the top row

Not a request, but a suggestion.  I would try to work in a picture from the full model earlier into the work.  When the reader sees the output of Figures 2/3, even for the xhuge models, it's not particularly overwhelming.  Once Figure 8 is presented, it's clear what the model is capable of.  I might move this forward (or make a smaller, related one), which captures what the model/approach is able to achieve.

**Strengths And Weaknesses:**

STRENGTHS:
* A method which enables more efficient training of diffusion models, both in terms of computation as well as data, has clear interest and value to the community
* Overall the text is clear and well written
* The related works does a good job highlighting the current approaches and how the proposed method differs/builds on them.
* The variations and ablation studies explored are well motivated and thorough.
* Use of, motivation for, and limitations of metrics are clear

WEAKNESS:
* The number of users involved in the evaluation is relatively small.  It would be good to significantly increase this to give the results more credence since this is the only comparative metric that exists.

---

> ### Author Response · Authors · 2024-08-05
> **The new version includes requested changes and presentation suggestions.**
>
> Thank you for your careful review of our paper and 's clear interest and value to the community. In the following we hope to address all your concerns one by one. If there is anything missing, please let us know.
> ___
> Requested Changes:
> We fixed the typos and quotes. Thanks for catching!
> In Figure 3, step 1.75 top row, the image generated a bright (close to white color) border, giving the impression of misalignment.
> We agreed with the reviewer and we  initially submitted figure 8 as a teaser image, but that was blocked by TMLR style. In this updated version we moved it into the Introduction section.
> ___
> Weakness:
>
> We kindly request the reviewer to review our response to reviewer Mr9y, as we address similar concerns with the human evaluation. In order to give it more credibility, we have also updated the paper to present generations Vermeer and SDXL side-by-side on the same prompts.

---

### Review · Reviewer_cYDt · 2024-07-22

**Summary Of Contributions:**

This paper proposes a way to implement super-resolution in diffusion models (text to image) through progressive growing strategies. The work is empirical, in that the algorithmic changes are at the network and training level. However, novelties in training strategy is important in complex setups such as diffusion models.
The training method is as follows. One first learns a 'core' components backbone that is common to all training setups. Following this, 'growing' is carried out for higher resolutions. The core is common to all trainings and skews, and does away with the cascading of multiple networks used generally in diffusion model training.

They train on the publicly available CC12M dataset. Following they present results on internal datasets trained with 8B parameters.

Evaluations are carried out to show the effectiveness of pieces involved - backbone size, greedy growing, evaluation metrics (e.g. FID, CLIP-score), prompt engineering and style tuning, diffusion guidance. They also show different tasks - e.g. counting, VQA.

**Audience:**

Yes

**Claims And Evidence:**

Yes

**Requested Changes:**

See above - It would be good to show what happens in the various diffusion steps.
A latent version of this work would be useful to examine. I see no reason why we cannot implement this idea at the latent level.
On guidance - perhaps it might be possible to optimize all the scores in a learned way?

**Strengths And Weaknesses:**

I find the general idea quite well motivated, and results convincing to see that the growing strategy works well. In addition, the authors have run many different experimental ablations to leave no doubt of its efficacy.

There are a few concerns, however. The underlying base model - what they call 'core' is a bit facile or obvious. They claim that it is analogous to the latent space of the VAE, but I find that an exaggeration. It looks like a generic extension of a pretrained backbone. Moreover, cascading or using hierarchical layers of pretrained backbone would only help (indeed, that is also done in latent versions of these models - e.g. hierarchical VAE [1]).

I also think we should examine the time dimension, what happens inside the various diffusion steps.

[1] https://arxiv.org/abs/2007.03898

---

> ### Author Response · Authors · 2024-08-08
> **Please clarify for our best understanding of your suggestions/requests**
>
> We thank the reviewer for their comments and ask for a few clarifications.
>
> Reviewer #cYDt mentions we claim that our strategy is analogous to the latent space of the VAE. We apologize if our report gives such an impression. We affirm we do not have the intention of claiming that as a contribution. We ask the reviewer to gently point out the elements in the text that are triggering that interpretation so that we can change them for better clarity.
>
> The reviewer also mentions we should “examine the time dimension, what happens inside the various diffusion steps.”. We kindly ask to clarify what exactly is being requested. We have not proposed changes to the diffusion process itself, nor to the time schedule. Our time schedule is identical to the Simple Diffusion paper (Hoogeboom et al., 2023b) and we claim no contribution in this direction.
>
> We agree with the reviewer in that we see no reason why our method cannot be extended to the latent models, and also that there is space for better optimizing guidance values. Both are considered out of the scope of this report and left as future work.

---

> ### Comment · Reviewer_cYDt · 2024-08-17
> **Clarifications**
>
> Time: I was looking for a picture on the lines of figure 3 in Hoogenboom et al, showing how the generations evolve over time. Thanks for clarifying the details about the scheduling process.
>
> Comment on latent model: Certain lines in the paper give the impression that the low dimensional representation (what the paper calls the 'core layers' is conceptually similar to the VAE's latent space. I now see though that the authors did not explicitly claim this, but were simply contrasting their approach to that of VQ-VAE based works. Specifically, here is an example (see below) from section 3.2. I retract this comment in the review as no explicit claim was made, although it does make the interpretation a bit more subtle.
>
> Excerpt from section 3.2
> "An alternative to the proposed use of the Shallow-UViT architecture, might be to train the core components.
> directly as an augmented ViT, as previously explored in latent diffusion models (Peebles & Xie, 2023). Our
> attempt to explore this approach proved not to be straightforward. A crucial difference between PSDM and
> LDM becomes highly relevant here. In the case of LDM, the transformer operates on latent tokens, and the
> diffusion model captures the latent token distribution. Our task, on the other hand, is to pretrain a rich
> representation directly from the raw pixels, for subsequent reuse as deep features within a higher-resolution
> pixel-space model. We conjecture that in such approaches the initial layers that are closer to the raw data
> do not transfer as well when reused within the final model."

---

### Decision · Action_Editor_CMVE · 2024-09-08

**Recommendation:** Accept with minor revision

**Comment:**

After the response, all the reviewers lean toward acceptance. This paper has some merits: the proposed growing strategy is well-motivated and works well; the experimental settings are thorough; the presentation is decent. The AC has read the paper and agrees with the opinions of the reviewers. However,  the paper still should undergo a minor revision, as the reviewers suggest the authors include more side-by-side comparisons against the current SotA methods (with the same prompts).

**Audience:**

The individuals who study generative models, particularly diffusion-based models, will be interested in the findings of this paper.

**Claims And Evidence:**

The claims in this work are well supported by the proposed experiments, which include comparisons with
other methods, ablation studies, and human evaluations.